# ViSAGe: Video-to-Spatial Audio Generation

**Jaeyeon Kim, Heeseung Yun & Gunhee Kim**
Seoul National University
jaeyeonkim99@snu.ac.kr, heeseung.yun@vision.snu.ac.kr, gunhee@snu.ac.kr

## Abstract

Spatial audio is essential for enhancing the immersiveness of audio-visual experiences, yet its production typically demands complex recording systems and specialized expertise. In this work, we address a novel problem of generating first-order ambisonics, a widely used spatial audio format, directly from silent videos. To support this task, we introduce YT-Ambigen, a dataset comprising 102K 5-second YouTube video clips paired with corresponding first-order ambisonics. We also propose new evaluation metrics to assess the spatial aspect of generated audio based on audio energy maps and saliency metrics. Furthermore, we present Video-to-Spatial Audio Generation (ViSAGe), an end-to-end framework that generates first-order ambisonics from silent video frames by leveraging CLIP visual features, autoregressive neural audio codec modeling with both directional and visual guidance. Experimental results demonstrate that ViSAGe produces plausible and coherent first-order ambisonics, outperforming two-stage approaches consisting of video-to-audio generation and audio spatialization. Qualitative examples further illustrate that ViSAGe generates temporally aligned high-quality spatial audio that adapts to viewpoint changes. Project page: https://jaeyeonkim99.github.io/visage

## 1 Introduction

Humans perceive the world through both auditory and visual cues, each of which conveys significant spatial information. Visual cues enable them to locate objects, while auditory cues help estimate the interaction betweeen objects in the environment based on the origin of sounds. Hence, spatial audio is vital for creating immersive experiences in the visual scenes (Poeschl et al., 2013; Holm et al., 2020; Hirway et al., 2022; Nguyen & Willson, 2023; Hirway et al., 2024). This makes spatial audio production essential for many applications such as film, virtual reality, and augmented reality. However, producing spatial audio typically requires expensive sound-field microphones, professional production equipment, and advanced technical expertise (Zotter & Frank, 2019).

Sound effects in videos are often recreated during the post-production stage due to challenges associated with on-location audio capture (Ament, 2014), adding more complexity to the task of producing spatial audio for visual content. Hence, generating appropriate spatial audio for silent videos has immediate and impactful applications for enhancing the immersive experience in various media. Moreover, recent advancements in generative models have enabled the creation of videos from textual descriptions (Ho et al., 2022b;a; Singer et al., 2023; Brooks et al., 2024), further increasing the demand for creating audio streams that align with spatial and contextual characteristics of videos.

Previous works have shown remarkable progress in generating audio from silent videos and spatial audio from mono audio. However, generating spatial audio directly from silent videos remains an unsolved challenge. Current video-to-audio generation models (Iashin & Rahtu, 2021; Luo et al., 2023; Wang et al., 2024; Pascual et al., 2024) are capable of producing audio based on the content and timing of the video. However, these models generate only mono audio, whereas spatial audio requires the generation of multiple channels with proper arrangement to convey a sense of space.

Audio spatialization models can generate binaural audio (Gao & Grauman, 2019; Zhou et al., 2020; Li et al., 2024b) or first-order ambisonics (Morgado et al., 2018; Lim & Nam, 2024), from mono audio using visual cues. However, they necessitate a reference mono audio, which is not available for silent videos. Combining video-to-audio generation and audio spatialization may introduce additional challenges. The generated audio often deviates from the ground-truth distribution and may not align with the timing or content of the video, potentially leading to inaccurate spatialization.

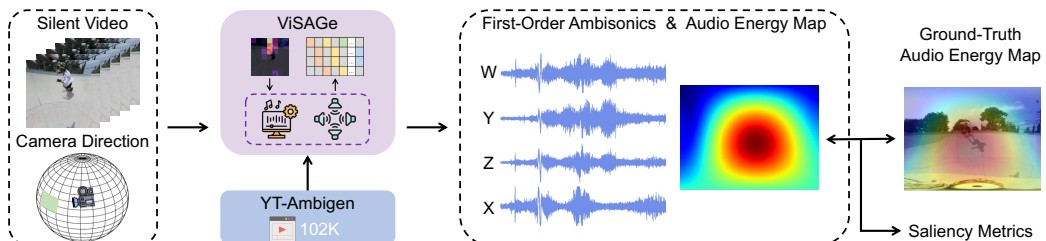

Figure 1: Video-to-Spatial Audio Generation. Given a silent video and the camera direction, the model generates corresponding first-order ambisonics. The camera direction gives cue about where the visual event occurs, enabling the model to generate an appropriate three-dimensional sound field.

In this work, we introduce a novel task: generating first-order ambisonics, a widely used spatial audio format, from silent videos, as in Figure 1. To address this task, we introduce YT-Ambigen, a new dataset comprising YouTube videos paired with first-order ambisonics, tailored for the audio generation. To evaluate the spatial quality of the generated ambisonics, we propose novel metrics derived from audio energy maps and visual saliency. Furthermore, we present Video-to-Spatial Audio Generation (ViSAGe), an end-to-end framework designed to generate and spatialize audio based on visual content and camera direction. ViSAGe leverages CLIP features, patchwise energy maps, and neural audio codecs along with rotation augmentation. Additionally, ViSAGe incorporates code generation scheme and guidance optimized for the simultaneous generation of multiple spatial channels. Extensive experiments on YT-Ambigen show that ViSAGe outperforms two-stage approaches, which separately handle video-to-audio generation and audio spatialization, across all metrics.

## 2 RELATED WORKS

**Video-to-Audio Generation**. Earlier works on creating soundtracks for silent videos focused on a limited set of classes (Owens et al., 2016; Zhou et al., 2018; Chen et al., 2020b). SpecVQGAN (Iashin & Rahtu, 2021) was the first to generate sounds for open-domain videos with autoregressive transformers trained on VQ-GAN (Esser et al., 2021) codebooks of mel-spectrograms. IM2WAV (Sheffer & Adi, 2023) extended this by training on hierarchical VQ-VAE (Razavi et al., 2019) codebooks. Diff-Foley (Luo et al., 2023) introduced contrastive audio-visual pretraining and latent diffusion for improved audio-visual synchronization, while V2A-mapper (Wang et al., 2024) used AudioLDM (Liu et al., 2023) by mapping CLIP (Radford et al., 2021) and CLAP (Wu et al., 2023).

The works most closely related to ours are FoleyGen (Mei et al., 2023) and MaskVAT (Pascual et al., 2024), which use CLIP features for visual conditioning and neural audio codecs (Défossez et al., 2023; Kumar et al., 2024) for audio generation. FoleyGen employs an autoregressive transformer, while MaskVAT uses masking-based generation (Chang et al., 2022). Unlike existing approaches that generate mono audio, ViSAGe generates multiple spatial channels simultaneously using spatial cues.

**Audio Spatialization with Visual Cues**. Generating spatial audio typically requires specialized equipment and expertise, motivating research into creating spatial audio from various conditions (Kushwaha et al., 2025; Heydari et al., 2025). In particular, several studies have explored generating binaural audio from mono audio using visual cues. Mono2Binaural (Gao & Grauman, 2019) proposed a UNet-like framework to predict binaural channel masks using visual cues. Sep-Stereo (Zhou et al., 2020) extended this by jointly modeling source separation and binauralization, while PseudoBinaural (Xu et al., 2021) applied the same approach to pseudo-binaural data generated from mono audio. More recent works utilize multitask-based geometry-aware features (Garg et al., 2023) and cyclic learning with localization (Li et al., 2024b) for binauralization.

Another line of works generate first-order ambisonics from mono audio based on panoramic videos. SpatialAudioGen (Morgado et al., 2018) produced spatial channels by separating sound sources inside the mono omnidirectional channel and localizing them in the correct direction. Similarly, Rana et al. (2019) predicted sound source locations and manually encoded the audio based on those predictions. Lim & Nam (2024) improved SpatialAudioGen by incorporating a pretrained source separation model and a channel panning loss between spatial channels. In this work, we focus on first-order ambisonics, a versatile spatial audio format that can be decoded into various formats, including binaural audio. Unlike the above methods, ViSAGe generates spatial audio purely from visual content, without the need for a reference mono audio.

**Audio Generation Using Neural Audio Codecs.** Neural audio codecs are autoencoders that compress audio signals into sequences of discrete codes (Zeghidour et al., 2021; Défossez et al., 2023; Kumar et al., 2024). Due to their discrete latent space and superior audio reconstruction, they are widely used for generating audio (Kreuk et al., 2023; Ziv et al., 2024), speech (Borsos et al., 2023; Wang et al., 2023), and music (Copet et al., 2023; Agostinelli et al., 2023; Ziv et al., 2024; Li et al., 2024a). Recent neural audio codecs often utilize residual vector quantization (RVQ), applying multiple codebooks to quantize the residuals from earlier steps (Zeghidour et al., 2021). This creates challenges in managing multiple code sequences, leading to various code generation strategies (Borsos et al., 2023; Wang et al., 2023; Agostinelli et al., 2023; Copet et al., 2023). While Copet et al. (2023) and Li et al. (2024a) explored strategies for stereo music, most works focus on mono audio due to the complexity of modeling residual code sequences even for a single channel. In this work, we propose an efficient method for generating all four channels of first-order ambisonics using neural audio codecs.

## 3 VIDEO-TO-AMBISONICS GENERATION

### 3.1 BACKGROUND: FIRST-ORDER AMBISONICS

Ambisonics is a three-dimensional surrounding sound format that captures and recreates sound fields using spherical harmonics. Due to its accurate and scalable representation of sound sources from all directions with desired precision, ambisonics plays a crucial role in immersive audio experiences. Among a variety of formats, First-order ambisonics (FOA) employs four channels $(W, X, Y, Z)$ to encode the sound field with first-order spherical harmonic decomposition. The $W$-channel corresponds to the sound from an omnidirectional microphone at the center, while each directional channel $(X, Y, Z)$ amounts to the sound from a figure-of-eight microphone aligned with the corresponding axis. FOA is more widely used than other higher-order representations due to its affordable and efficient nature as well as compatibility with popular video streaming services like YouTube. Compared to other surround sound formats like 5.1 surround sound, which favors a fixed frontal field, ambisonics offers unbiased playback of directional information with precision (Courville & Studio, 1994), promoting immersiveness in dynamic user-centric scenarios.

One of the primary advantages of ambisonics format is that we can explicitly map the energy of auditory information to the spherical coordinate system. This energy map reveals the direction from which the audio energy originates, representing the spatial characteristics of the sound. Using spherical harmonics decomposition, we can derive an audio energy map $G(\phi, \theta)$ for $\phi \in [0, \pi], \theta \in [0, 2\pi]$ with respect to real spherical harmonics $\mathbf{Y}_l^m(\phi, \theta)$ and audio length $L$:

$$G(\phi, \theta) = \frac{1}{L} \sum_{t=1}^{L} \left( \mathbf{Y}_0^0(\phi, \theta)W(t) + \mathbf{Y}_{-1}^1(\phi, \theta)Y(t) + \mathbf{Y}_0^1(\phi, \theta)Z(t) + \mathbf{Y}_1^1(\phi, \theta)X(t) \right). \quad (1)$$

### 3.2 TASK DESCRIPTION

Video-to-ambisonics generation addresses the problem of generating FOA channels $(W, X, Y, Z)$ given silent video frames. This brings about major challenges in methodology and evaluation that have not discussed in prior spatialization or video-to-audio generation tasks. Compared to mono audio generation problems, each generated channel should be plausible and visually entailing while maintaining consistency with one another. Moreover, all channels should form a spatially coherent sound field for immersiveness, *i.e.*, the audio perceived by a person viewing from a specific direction should also be plausible. Such spatial coherency should hold no matter what content is conveyed in the generated audio by preserving the directions of dominant sound sources.

For visual conditioning, we use a combination of field-of-view (FoV) videos and their corresponding camera directions. Although ambisonics are typically provided with panoramic videos due to their three-dimensional nature (Morgado et al., 2018; 2020), we opt for FoV-based conditioning since FoV has much broader applications compared to panoramic videos and can be easily integrated with traditional video-to-audio generation methods and benchmarks. However, FoV alone lacks information about where visual events occur within the three-dimensional environment. To address this, we include the camera direction as input, representing where the visual scene is taking place, allowing the model to create an accurate sound field based on the visual content. This approach not only enhances spatial awareness but also offers users greater control when generating the FOA.

### 3.3 EVALUATION METRICS

**Semantic Metrics.** We adopt two widely used metrics, Fréchet Audio Distance (FAD) (Roblek et al., 2019) and Kullback-Leibler Divergence (KLD), to evaluate the semantic aspects of the generated audio. FAD is defined as the Fréchet distance between the feature distributions of the generated and ground-truth audio, as extracted by a pretrained audio encoder. FAD reflects the perceptual quality and fidelity of the generated audio, whereas KLD measures the KL divergence between the class distributions of the generated and ground-truth audio, evaluating how well the generated audio captures the intended audio concepts. Since the pretrained classifiers used for metric computation require mono audio input, we decode both the ground-truth and predicted FOA into mono audio based on the ground-truth camera direction $(\phi, \theta)$, where $\phi$ represents the azimuth and $\theta$ the elevation:

$$s(\phi, \theta) = W + X \cos \phi \cos \theta + Y \sin \phi \cos \theta + Z \sin \theta. \tag{2}$$

$W, X, Y$, and $Z$ represent the respective channels of the FOA. The decoded mono audio is equivalent to a recording from the virtual 3D cardioid microphone heading $(\phi, \theta)$, reflecting what the listener would likely hear in the scene. We use the decoded mono audio, rather than $W$, as the representative mono audio to ensure that the semantic coherence of the generated ambisonics channels can be evaluated. We report FAD and KLD evaluated on decoded mono audio, *i.e.,* $\textbf{FAD}_{\textbf{dec}}$ and $\textbf{KLD}_{\textbf{dec}}$.

For generated FOA, the fidelity of each channel is also crucial to the listener's experience, since these channels can be combined in various ways depending on the listener's location and direction. To assess the fidelity of the individual channels, we report the average of FAD from each channel, *i.e.,* $\textbf{FAD}_{\textbf{avg}}$, which can capture the overall plausibility of generated ambisonics.

**Spatial Metrics.** Previous works on audio spatialization (Morgado et al., 2018; Gao & Grauman, 2019) have utilized distance-based metrics such as STFT, Log-spectral, and Envelope distance, which compare spatialized channels from reference mono audio with ground-truth spatial channels. However, these metrics are not suitable for evaluating generated audio with varying content and timing, since they cannot be directly compared to ground-truth audio. To address this limitation, we propose a new set of metrics to evaluate the spatial aspects of generated ambisonics.

As in Eq. 1, first-order ambisonics can be used to generate an audio energy map over the sphere using spherical harmonics decomposition. We adopt visual saliency metrics (Bylinskii et al., 2018) to evaluate the similarity between the original energy map and the generated energy map, typically in the form of a heatmap over an equirectangular panorama of elevation by azimuth (Cheng et al., 2018). A key distinction in this adaptation is that we mitigate oversampling bias in evaluating saliency. The energy evaluation between equirectangular panoramas is prone to oversampling around $\theta = 0$ and $\theta = \pi$, making the evaluation less accurate when the auditory source deviates from the center. We prevent this with trivial overhead by reducing the number of sampled points for evaluation by $\sin \theta$.

We calculate the Correlation coefficient (CC) and the Area Under the Curve (AUC) values between the audio energy maps of the generated ambisonics and the ground-truth audio. To measure the spatial coherence of the generated sound field with respect to varying temporal granularity, we report CC and AUC for different temporal windows: energy map over full generated audio ($\textbf{CC}_{\textbf{all}}$, $\textbf{AUC}_{\textbf{all}}$), energy map aggregated every 1000ms ($\textbf{CC}_{\textbf{1fps}}$, $\textbf{AUC}_{\textbf{1fps}}$) and 200ms ($\textbf{CC}_{\textbf{5fps}}$, $\textbf{AUC}_{\textbf{5fps}}$).

## 4 DATASET: YT-AMBIGEN

**Motivation.** Existing datasets on video-to-audio generation or spatialization fall short in addressing the video-to-ambisonics generation. As outlined in Table 1, only a restricted number of datasets cover video-to-audio problems at scale (*i.e.*, >100h). Previous datasets with spatial audio either only contain 360° videos as visual conditions or have not been demonstrated in audio generation.

In addition, video-to-audio generation itself is considerably challenging, even with the largest ambisonics video dataset available. Experimental results in Table 2 suggest that a competitive generative model on VGGSound (Chen et al., 2020a) struggles to train or finetune with YT360 (Morgado et al., 2020), often producing noise-like sounds as outputs. We hypothesize this performance gap largely attributes to the quality of existing datasets for generative problems. Therefore, we newly propose a large-scale dataset specifically designed to meet the needs of video-to-ambisonics generation, enabling more accurate and contextually relevant spatial audio synthesis. **YT-Ambigen** dataset com-

Table 1: Comparison of YT-Ambigen with existing datasets. FoV and 360° respectively denote field-of-view videos and panoramic videos. NS, B, and FOA refer to non-spatial audio, binaural audio, and first-order ambisonics, respectively. (*Number of audio classes < 15)

| Dataset | # of Clips | Video Length | Video Type | Audio Type | All Spatial Channels | Open Domain | Audio Generation |
|---|---|---|---|---|---|---|---|
| Greatest Hits (Owens et al., 2016) | 1K | 11h | FoV | NS | - | ✗ | ✓ |
| VEGAS (Zhou et al., 2018) | 28K | 55h | FoV | NS | - | ✓* | ✓ |
| VAS (Chen et al., 2020b) | 13K | 24h | FoV | NS | - | ✓* | ✓ |
| VGGSound (Chen et al., 2020a) | 200K | 560h | FoV | NS | - | ✓ | ✓ |
| FairPlay (Gao & Grauman, 2019) | 2K | 5h | FoV | B | ✓ | ✗ | ✗ |
| OAP (Vasudevan et al., 2020) | 64K | 26h | 360° | B | ✓ | ✗ | ✗ |
| REC-STEEET (Morgado et al., 2018) | 123K | 3.5h | 360° | FOA | ✓ | ✗ | ✗ |
| YT-ALL (Morgado et al., 2018) | 3976K | 113h | 360° | FOA | ✗ | ✓ | ✗ |
| YT-360 (Morgado et al., 2020) | 89K | 246h | 360° | FOA | ✗ | ✓ | ✗ |
| STARSS23 (Shimada et al., 2024) | 0.2K | 7.5h | 360° | FOA | ✓ | ✗ | ✗ |
| **YT-Ambigen** | 102K | 142h | FoV | FOA | ✓ | ✓ | ✓ |

prises a total of 102,364 five-second FoV clips with corresponding FOA and camera direction $(\phi, \theta)$, which is divided into 81,594 / 9,604 / 11,166 clips for training, validation, and test, respectively.

**Dataset Curation.** We address two major issues identified in YT360 for video-to-audio generation: (i) the absence of semantically significant audio events due to amplitude-based filtering and (ii) weak coherence between the audio events and visual information. Using 5.2K panoramic videos with first-order ambisonics collected from YouTube, we first filter out videos where the average absolute amplitude of any channel per second is less than $10^{-20}$ to ensure the presence of all four channels.

Table 2: Video-to-audio generation results of the model in Sec. 5 for different datasets. $\mathcal{X} \to \mathcal{Y}$ indicates pretraining and finetuning datasets.

| Dataset | FAD↓ | KLD↓ |
|---|---|---|
| VGGSound | 3.62 | 2.23 |
| YT360 | 15.91 | 3.10 |
| VGGSound → YT360 | 12.92 | 3.16 |
| YT-Ambigen | 3.95 | 1.77 |
| VGGSound → YT-Ambigen | 3.42 | 1.75 |

We then focus on clips with noticeable audio events that are suitable for generation. For each 1s segment in all videos, we determine the validity of the clip by thresholding the root mean square energy of the segment. These segments are merged into 5s clips by retaining only those with >3s of valid audio. We utilize a 5s temporal window to capture coherent events with sufficient length for generation, considering longer clips tend to include fewer valid segments. To further ensure each clip contains semantically recognizable audio events, we use an AudioSet classification model (Koutini et al., 2022) to recursively select clips with high-probability sound event labels. Selected clips cover over 300 distinct AudioSet classes, demonstrating their suitability for open-domain generation.

Moreover, we try to ensure that the visual cues for audio generation are within the FoV. We calculate the audio energy map for each clip to identify the argmax coordinate, where the sounding events are likely happening. We then crop the panoramic video around this point to obtain FoV videos. Finally, we filter out clips based on audio-visual relevance scores (Luo et al., 2023), removing any clips with scores lower than one standard deviation below the mean.

## 5 APPROACH: VISAGE

The overall architecture of the proposed model, ViSAGe, is illustrated in Figure 2-(a). Let video frames be $V \in \mathbb{R}^{T \times 3 \times H \times W}$, where $T$, $H$, and $W$ denote the time, height, and width, respectively. The camera direction is given by $D = (\phi, \theta)$ for azimuth $\phi$ and elevation $\theta$. The goal of ViSAGe is to generate first-order ambisonics $A = (W, X, Y, Z) \in \mathbb{R}^{4 \times L}$, where $L$ is the length of the waveform, by modeling the conditional probability $p(A|V, D)$ with transformer encoder-decoder architecture.

### 5.1 CONDITIONAL ENCODING

The video frames $V$ and the camera direction $D$ is conditioned through the transformer encoder. For $V$, we use CLIP (Radford et al., 2021) features to capture semantic content, while proposing the use of patchwise energy maps to capture fine-grained spatial cues within the frames. Meanwhile, $D$ is processed into a direction embedding to provide cues for overall spatiality.

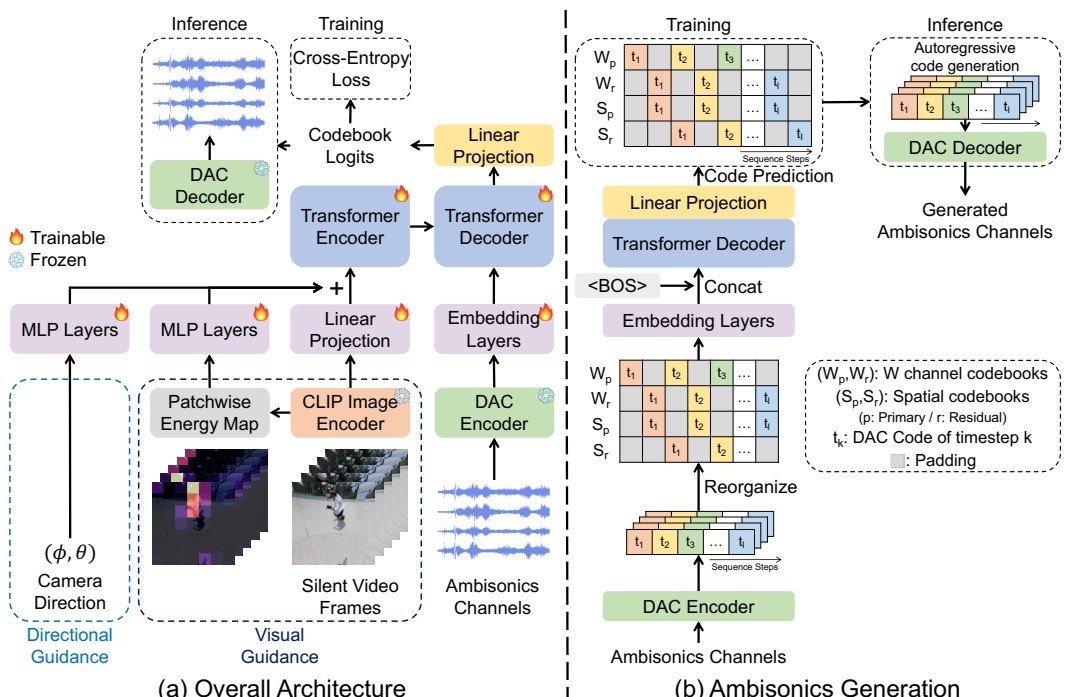

Figure 2: (a) Overall architecture of ViSAGe and (b) its ambisonics generation with DAC codes. Each block in (b) represents all residual codes belonging to the corresponding codebook group.

**CLIP Embeddings.** We choose CLIP over other visual encoders due to its superior performance in previous video-to-audio generation works (Sheffer & Adi, 2023; Mei et al., 2023; Pascual et al., 2024). The video frames $V \in \mathbb{R}^{T \times 3 \times H \times W}$ are transformed using a pretrained CLIP image encoder and joint projection, and then linearly projected to align with the transformer's hidden dimension, yielding $I \in \mathbb{R}^{T \times d_t}$, where $d_t$ is the dimension of the transformer encoder's hidden states.

**Patchwise Energy Maps.** While CLIP effectively captures the overall semantics of visual frames, it lacks spatial information, such as the location and movement of sounding objects. To address this limitation, we introduce using a patchwise energy map as an additional visual input to extract fine-grained, temporally aligned spatial information from video frames. First, we obtain patch-level image embeddings before pooling, denoted as $e_p \in \mathbb{R}^{T \times h \times w \times d_p}$, from the pretrained CLIP image encoder. Here, $h = H/p$ and $w = W/p$ with the patch size $p$, and $d_p$ is the dimension of the CLIP image embeddings before the joint projection. Similar to the spatial and temporal saliency approach from PAVER (Yun et al., 2022), the spatial and temporal scores of each patch are calculated based on the patch's embedding distance from its spatial and temporal neighbors. Let $x_{ij}^t \in \mathbb{R}^{d_p}$ denote the embedding for the $ij$-th patch at time-step $t$. The spatial score $S_{ij}^t$ and the temporal score $T_{ij}^t$ for embedding $x_{ij}^t$ given spatial and temporal window $N, T$ and cosine similarity $d_s$ are defined as

$$S_{ij}^t = 2 - 2d_s(x_{ij}^t, \frac{1}{(2N+1)^2} \sum_{k=i-N}^{i+N} \sum_{l=j-N}^{j+N} x_{kl}^t), \quad T_{ij}^t = 2 - 2d_s(x_{ij}^t, \frac{1}{2T+1} \sum_{k=t-T}^{t+T} x_{ij}^k). \quad (3)$$

High spatial scores indicate that a patch contains content distinct from adjacent patches, while high temporal score suggests that patch contains temporally changing information such as a moving object. Therefore, these scores are correlated with the location and movement of the sounding object in the scene. Next, the scores are converted into probabilities by applying softmax over the patches and averaged, forming an energy map over the patches, *i.e.*, $E \in [0, 1]^{T \times h \times w}$. This energy map is flattened and passed through MLP layers, resulting in the final energy map embedding $I_e \in \mathbb{R}^{T \times d_t}$.

**Direction Embedding.** We use the direction embedding to control the overall spatial directivity of the sound field. The camera direction $D = (\phi, \theta)$ is first mapped to a unit vector $u \in \mathbb{R}^3$ in Cartesian coordinates for smooth interpolation across different directions. This unit vector is then projected through MLP layers and duplicated along the $T$ axis to form the direction embedding $I_d \in \mathbb{R}^{T \times d_t}$.

**Transformer Encoder.** To condition the input features, the embeddings $I$, $I_e$, and $I_d$ are summed and then concatenated with learnable embeddings that represent the start and end of the sequence along the temporal dimension. Positional embeddings are added to capture the sequential order of the inputs. The resulting features are fed into the transformer encoder layers.

## 5.2 AMBISONICS GENERATION

We model first-order ambisonics using a neural audio codec, which encodes waveforms into a sequence of discrete codes. This enables the use of discrete modeling techniques such as autoregressive generation, while the predicted codes can be decoded back into waveforms. To facilitate this process, we propose strategies to model neural audio codes for ambisonics generation.

**Descript Audio Codec Encoding.** The FOA channels are transformed into an audio code matrix using the Descript Audio Codec (DAC) encoder (Kumar et al., 2024), a state-of-the art nerual codec for open-domain audios (Wu et al., 2024). Based on residual vector quantization (RVQ) that quantizes the residuals of previous codebooks, DAC encoder compresses each ambisonics channel into a discrete code matrix $C \in \mathbb{V}^{N \times L_c}$, where $N$ is the number of codebooks used in the RVQ process, $L_c$ is the length of the compressed audio, and $\mathbb{V}$ is the vocabulary of the codebooks. Each row $C_{n,:}$ corresponds to the code sequence from a specific codebook, while each column $C_{:,t}$ represents the codes at a given time step. To handle all channels simultaneously, we concatenate the code matrices from all four ambisonics channels along the codebook dimension, forming $C_a \in \mathbb{V}^{4N \times L_c}$.

**The Code Generation Pattern.** The proposed code generation pattern is illustrated in Figure 2-(b). Generating first-order ambisonics channels requires handling four times more code sequences compared to mono audio while ensuring both semantic and temporal coherence across all channels. For a given ambisonics code matrix $C_a \in \mathbb{V}^{4N \times L_c}$ and codebook index $1 \leq i \leq 4N$, we divide them into four groups to effectively model the dependencies between code sequences: $W_p$, $W_r$, $S_p$, and $S_r$. $W$ and $S$ denotes omnidirectional ($W$) and spatial ($X, Y, Z$) channels and $p$ and $r$ denotes primary and residual codebooks, respectively. That is, for $\mathcal{P} = \{i \mid i \bmod N = 1\}$ and $\mathcal{W} = \{i \mid 1 \leq i \leq N\}$, $W_p = \{(C_a)_{i,:} \mid i \in \mathcal{P}, i \in \mathcal{W}\}$, $W_r = \{(C_a)_{i,:} \mid i \notin \mathcal{P}, i \in \mathcal{W}\}$, $S_p = \{(C_a)_{i,:} \mid i \in \mathcal{P}, i \notin \mathcal{W}\}$, and $S_r = \{(C_a)_{i,:} \mid i \notin \mathcal{P}, i \notin \mathcal{W}\}$.

We hypothesize that two types of dependencies should be modeled among these groups of codebooks. First, we must capture the dependency between the primary and residual codebooks (residual dependency), as later codebooks depend on earlier ones in RVQ. Second, we need to model the dependency between the omnidirectional and spatial codebooks (spatial dependency), since the spatial channels should remain semantically coherent with the omnidirectional channel while varying in amplitude according to spatial cues. A naive approach would model these dependencies sequentially in the order of $W_p \rightarrow W_r \rightarrow S_p \rightarrow S_r$; yet, it results in a sequence length of $4L_c$.

To address this, we propose a more efficient generation pattern that requires only $2L_c + 1$ steps. We first generate $W_p$, followed by $(W_r, S_p)$, then $(W_p, S_r)$, and so forth. For sequential step $1 \leq s \leq 2L_c + 1$ and codebook index $1 \leq i \leq 4N$, we modify $C_a$ to $C'_a \in \mathbb{V}^{4N \times (2L_c+1)}$:

$$(C'_a)_{i,s} = \begin{cases} (C_a)_{i, \frac{s+1}{2}} & \text{if } s \bmod 2 = 1, \ s \neq 2L_c + 1, \ (C_a)_{i,:} \in W_p \\ (C_a)_{i, \frac{s-1}{2}} & \text{if } s \bmod 2 = 1, \ s \neq 0, \ (C_a)_{i,:} \in S_r \\ (C_a)_{i, \frac{s}{2}} & \text{if } s \bmod 2 = 0, \ (C_a)_{i,:} \in W_r \cup S_p \\ \varnothing & \text{else (padding)} \end{cases} \tag{4}$$

Autoregressivley modeling the sequential columns $(C'_a)_{:,s}$ enables modeling both residual and spatial dependencies. $C'_a$ serves as both the input and the prediction target for the transformer decoder.

**Transformer Decoder.** In order to model discrete code matrix $C'_a$ with the transformer decoder, each row is embedded using separate embedding layers and then summed with positional embeddings. A learnnable <BOS> embedding is concatenated at the start of the sequence to be used for autoregressive generation. The resulting sequence is passed through the transformer decoder layers. The final hidden states of the decoder are fed into separate linear layers, which predict the logits for each row of $C'_a$.

**Training and Inference.** During training, we use the cross-entropy loss between the ground-truth code matrix $C'_a$ and prediction. During inference, the model autoregressively generates codes for each sequential step. These codes are reorganized to the original DAC code matrix format, and decoded through the DAC decoder to generate respective channels. More details are deferred to the Appendix.

**Rotation Augmentation.** To guide the model in capturing spatial aspects and to disentangle visual features from the viewing direction, we utilize a rotation augmentation strategy. For a given rotation matrix $R \in \mathbb{R}^{3 \times 3}$, the sound field of the first-order ambisonics channels $(W, X, Y, Z)$ can be rotated as $W' = W$ and $(X', Y', Z')^T = R(X, Y, Z)^T$, where $(W', X', Y', Z')$ denotes the rotated ambisonics channels. Since elevation is closely tied to visual features—e.g., the sound of a river cannot originate from above if the river is flowing below—we perform azimuth rotation for augmentation. Specifically, during training, with a probability of 0.5, we rotate the azimuth by $90°$ along the $z$-axis, while simultaneously adjusting the input viewing direction from $D = (\phi, \theta)$ to $D' = (\phi + \frac{\pi}{2}, \theta)$. This rotation is selected for augmentation because it is computationally efficient and can be implemented by $(W', X', Y', Z') = (W, -Y, X, Z)$, while minimizing the loss of information that could arise from altering the relationship between direction and visual features.

**Directional and Visual Guidance.** We employ classifier-free guidance (Ho & Salimans, 2022) on the predicted logits for DAC codes to enhance the generation of spatial audio. We introduce applying classifier-free guidance to the directional condition to improve the spatial accuracy of the audio. During training, the directional unit vectors $u$ are replaced with null embeddings with the probability of 0.1. During inference, unconditional logits for classifier-free guidance are generated by replacing $u$ with null embeddings. For the reorganized code matrix $C'_a \in \mathbb{V}^{4N \times (2L_c+1)}$, let $C'_t = (C'_a)_{:,t}$. Given input video frames $V$ and a camera direction $D$, DAC codes for sequence step $t$ are sampled from

$$\log p_\phi(C'_t \mid C'_{1:t-1}, V, D) + \omega(\log p_\phi(C'_t \mid C'_{1:t-1}, \varnothing, D) - \log p_\phi(C'_t \mid C'_{1:t-1}, \varnothing, \varnothing)), \quad (5)$$

where $\omega$ and $p_\phi$ denotes the guidance scale and the probability parameterized by ViSAGe, respectively.

Furthermore, we adopt guiding both the directional and visual conditions simultaneously to further improve the semantic quality of the generated audio. CLIP embeddings and the patch-wise energy map $E$ are additionally replaced with null embeddings with a probability of 0.1 during training. At inference time, we guide both conditions jointly, as we observe that they are closely related. Thus, DAC codes are sampled from the modified log probability that replaces the first $\varnothing$ to $V$. Based on a hyperparameter sweep, we use guidance scale $\omega = 2.5$ throughout the experiments.

## 6 EXPERIMENT

### 6.1 SETUP

We use YT-Ambigen as a main testbed for video-to-ambisonics generation with the evaluation protocol explained in Sec. 3.3. We first pretrain ViSAGe on VGGSound for mono audio generation, followed by finetuning on YT-Ambigen. During pretraining, only CLIP features are used as input, and we train the codebook embeddings and projection layers corresponding to the W-channel.

**Baselines.** Since there is no prior work to directly generate first-order ambisonics from FoV videos like ViSAGe, we compose baselines by merging video-to-audio generation with audio spatialization. For video-to-audio generation, we adopt SpecVQGAN (Iashin & Rahtu, 2021) and Diff-Foley (Luo et al., 2023) as state-of-the-art open-domain generation models with publicly available implementation. We finetune the models pretrained on VGGSound to generate W-channel audio for YT-Ambigen.

We employ two methods to spatialize the W-channel audio generated by the video-to-audio models. First, we encode first-order ambisonics based on the ground-truth direction (Ambi Enc.). We encode a mono sound source $s(t)$ at direction $(\phi, \theta)$, into first-order ambisonics as follows (Zotter & Frank, 2019): $W = \frac{1}{\sqrt{2}}s(t)$, $X = \cos\phi\cos\theta s(t)$, $Y = \sin\phi\cos\theta s(t)$, and $Z = \sin\theta s(t)$. Spatial audio workstations typically encode mono audio into ambisonics by applying the eqution to each sound source composing the mono signal. As a straightforward spatialization approach, we manually encode W-channel to FOA by treating $s(t) = \sqrt{2}W$. This is conceptually equivalent to encoding the generated sound from a speaker located at $(\phi, \theta)$ into first-order ambisonics.

Additionally, we train an audio spatialization model based on visual cues (Audio Spatial.). Since none of the previous methods are fully compatible with our current setup, we train a spatialization model from scratch as a baseline. By closely following the architectures in Garg et al. (2023) and Liu et al. (2024), the spatializer consists of a U-Net architecture where the model learns to predict directional audio $(X, Y, Z)$ from the complex spectrogram of the $W$-channel. We tile and concatenate the visual features to the output of the audio encoder and decode to train with L2 loss. We use CLIP as the visual feature backbone and encode camera direction for a fair comparison.

Table 3: Results on YT-Ambigen. PT, DIR, PE, and RA stand for pretraining, directional embedding, patchwise energy map, and rotation augmentation, respectively.

| Model | | Semantic Metrics | | | Spatial Metrics | | | | | |
|---|---|---|---|---|---|---|---|---|---|---|
| | | | | | | CC↑ | | | AUC↑ | |
| V2A | Spatialization | $FAD_{dec}\downarrow$ | $KLD_{dec}\downarrow$ | $FAD_{avg}\downarrow$ | All | 1fps | 5fps | All | 1fps | 5fps |
| **Comparison to baseline models** | | | | | | | | | | |
| SpecVQGAN | Ambi Enc. | 5.94 | 2.56 | 5.62 | 0.349 | 0.337 | 0.322 | 0.687 | 0.680 | 0.670 |
| | Audio Spatial. | 6.40 | 2.43 | 7.90 | 0.619 | 0.587 | 0.547 | 0.848 | 0.828 | 0.802 |
| Diff-foley | Ambi Enc. | 5.68 | 2.60 | 5.53 | 0.349 | 0.337 | 0.322 | 0.687 | 0.680 | 0.670 |
| | Audio Spatial. | 7.24 | 2.51 | 8.76 | 0.577 | 0.537 | 0.494 | 0.826 | 0.803 | 0.777 |
| ViSAGe (Directional) | | 5.56 | 2.01 | 4.76 | **0.721** | **0.671** | **0.624** | **0.890** | **0.864** | **0.839** |
| ViSAGe (Directional & Visual) | | **3.86** | **1.71** | **4.20** | 0.635 | 0.584 | 0.531 | 0.846 | 0.819 | 0.790 |
| **Ablation on Model Components** | | | | | | | | | | |
| PT | DIR | PE | RA | | | | | | | |
| ✓ | | | | **3.73** | 1.76 | 4.19 | 0.430 | 0.398 | 0.362 | 0.724 | 0.708 | 0.689 |
| ✓ | ✓ | | | 3.74 | 1.77 | **4.04** | 0.524 | 0.482 | 0.439 | 0.778 | 0.757 | 0.734 |
| | ✓ | ✓ | | 4.44 | 1.89 | 4.49 | 0.544 | 0.498 | 0.449 | 0.794 | 0.770 | 0.745 |
| ✓ | ✓ | ✓ | | 4.01 | 1.78 | 4.30 | 0.531 | 0.486 | 0.441 | 0.782 | 0.759 | 0.735 |
| ✓ | ✓ | ✓ | ✓ | 3.86 | **1.71** | 4.20 | **0.635** | **0.584** | **0.531** | **0.846** | **0.819** | **0.790** |
| **Ablation on Code Generation Pattern** | | | | | | | | | | |
| Sequential Delay | | 10.75 | 2.66 | 8.66 | 0.386 | 0.341 | 0.311 | 0.746 | 0.716 | 0.695 |
| Residual Only | | 4.68 | 1.97 | 4.37 | 0.480 | 0.448 | 0.413 | 0.777 | 0.759 | 0.738 |
| Spatial Only | | 5.89 | 2.01 | 5.69 | 0.511 | 0.468 | 0.425 | 0.779 | 0.756 | 0.732 |
| Ours | | **4.44** | **1.89** | **4.49** | **0.544** | **0.498** | **0.449** | **0.794** | **0.770** | **0.745** |

## 6.2 RESULTS

Table 3 presents the overall results. ViSAGe with directional guidance outperforms the two-stage baselines in both semantic and spatial metrics, demonstrating its capability to generate semantically rich and spatially coherent first-order ambisonics. Importantly, manually encoded FOA exhibits better semantic quality but fail to capture spatial aspects adequately. On the other hand, ambisonics produced via the spatialization model effectively capture spatial information but suffer from reduced audio fidelity. In contrast, ViSAGe successfully balances semantic and spatial aspects, generating semantically and spatially accurate audio. Additionally, ViSAGe with both visual and directional guidance further improves semantic quality, albeit with some degradation in spatial accuracy. Nevertheless, it performs comparably to the best-performing two-stage approach in spatial metrics, while significantly outperforming it in semantic metrics. In subsequent experiments, we use both guidance.

**Ablation on Model Components.** We conduct ablation studies on several key components, including pretraining on VGGSound, direction embedding, patchwise energy maps, and rotation augmentation. The results in Table 3 show that while the overall semantic quality remains consistent when pretrained on VGGSound, both direction embedding and rotation augmentation significantly enhance spatial accuracy. Patchwise energy maps also contribute to increased spatial metrics by capturing fine-grained spatial details in the FoV scenes. Moreover, pretraining on diverse video clips from VGGSound helps the model to generate semantically plausible audio.

**Ablation on Code Generation Pattern.** We compare the proposed code generation pattern with alternative patterns that generates ambisonics channels in a similar or shorter number of steps. Let $C_a \in V^{4N \times L_c}$ be the ambisonics code matrix. First, we adopt the sequential delay pattern (Li et al., 2024a) from stereo music generation, which delays the generation of later codebooks to condition them on earlier ones, requiring $L_c + 4N - 1$ steps. We also compare our method with patterns that model only residual dependency, following $(W_p, S_p) \to (W_r, S_r)$, and only spatial dependency, using $(W_p, W_r) \to (S_p, S_r)$, both requiring $2L_c$ steps. Patterns are illustrated in Figure 4.

The results in Table 3 show that the sequential delay pattern fails to model both semantic and spatial aspects, indicating that generating first-order ambisonics cannot be achieved by simply adopting patterns that are successful for mono or stereo audio generation. Additionally, modeling only residual dependency improves semantic quality compared to modeling only spatial dependency, but performs worse in capturing spatial accuracy. This suggests that residual dependency is more closely linked to semantic quality, while spatial dependency is critical for spatial accuracy. In contrast, our proposed code generation pattern outperforms all other patterns in both semantic and spatial metrics, effectively modeling both dependencies while maintaining a similar number of steps.

**Qualitative Examples.** As illustrated in Figure 3, the linear spectrograms demonstrate that ViSAGe generates semantically and spatially coherent ambisonics channels. We also compare the audio

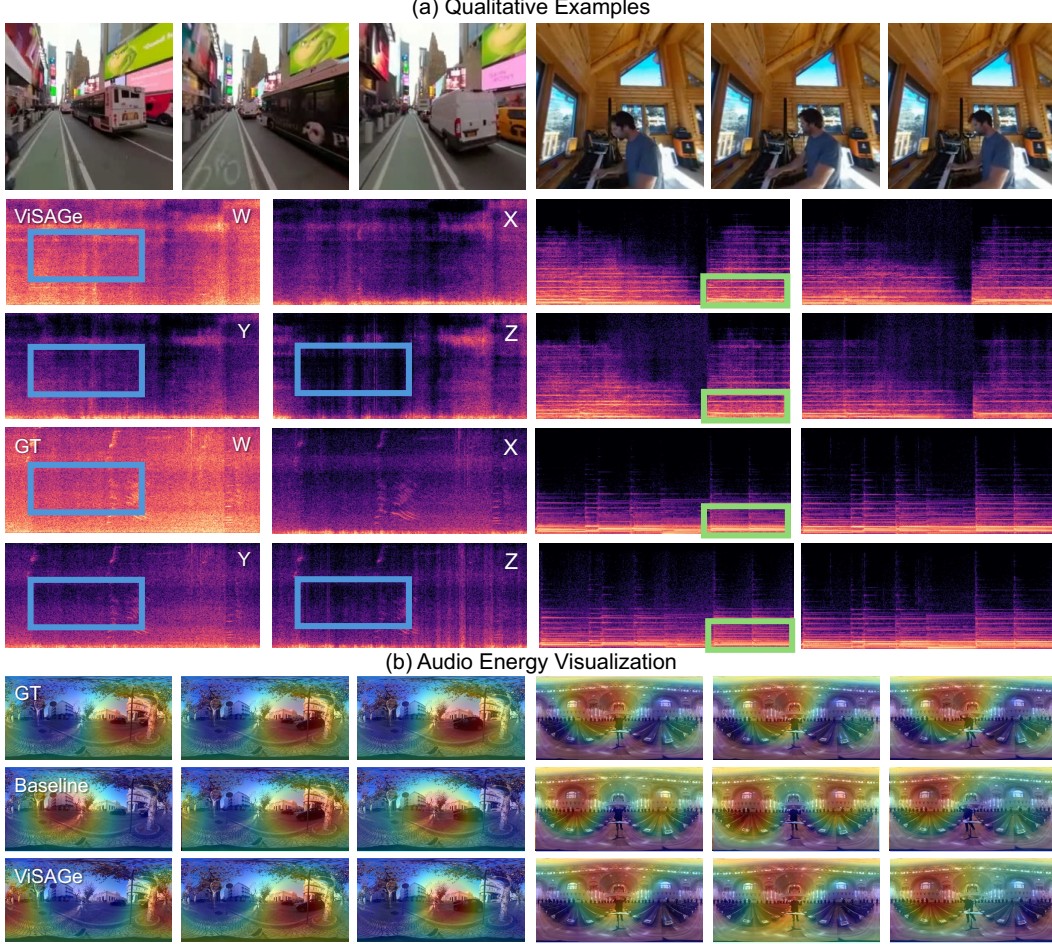

Figure 3: (a) Qualitative examples of generated audios and (b) audio energy visualization. Blue boxes highlight ViSAGe's ability to capture differences between spatial channels, while green boxes demonstrate that ViSAGe generates semantically plausible events.

energy maps from ViSAGe with those from the ablated model, which is conditioned only on CLIP features. While the ablated model captures some degree of spatiality due to the inherent relationship between visual features and spatial aspects, ViSAGe captures significantly more fine-grained details and temporal dynamics, highlighting the effectiveness of the proposed components. Additional qualitative examples are provided in Appendix F.

## 7 CONCLUSION

We addressed the challenging task of generating spatial audio, specifically first-order ambisonics, directly from silent videos—a task that has significant implications for enhancing the realism and immersiveness of audio-visual media. We introduced YT-Ambigen as a large-scale dataset that pairs YouTube video clips with corresponding first-order ambisonics, providing a valuable resource for future research in this area. To rigorously assess the spatial fidelity of generated audio, we proposed novel metrics that incorporate audio energy maps and visual saliency. Our proposed framework, ViSAGe, uniquely integrates neural audio codecs with CLIP-derived visual features, enabling the generation of semantically rich and spatially coherent ambisonics from video frames only. Experimental results confirmed that ViSAGe outperforms two-stage methods in both semantic and spatial evaluations. The promising performance of ViSAGe, along with its ability to adapt to dynamic visual contexts, underscores its potential for broad application in immersive media production. Future work will explore further enhancements in spatial audio realism and extend the framework's applicability to other forms like higher-order ambisonics.

**Acknowledgement.** This work was supported by Institute of Information & Communications Technology Planning & Evaluation (IITP) grant (No. RS-2019-II191082, SW StarLab; No. RS-2021-II211343, Artificial Intelligence Graduate School Program (Seoul National University; No. RS-2022-II220156, Fundamental research on continual meta-learning for quality enhancement of casual videos and their 3D metaverse transformation) the National Research Foundation of Korea (NRF) grant (No. 2023R1A2C2005573) funded by the Korea government (MSIT). Gunhee Kim is the corresponding author.

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

Table 4: Ablation on Classifier-Free Guidance. For Dual $\omega_1$ & $\omega_2$, $\omega_1$ denotes the guidance scale for the directional guidance, while $\omega_2$ denotes the guidance scale for the visual guidance.

| Guidance Scheme | Semantic Metrics | | | Spatial Metrics | | | | | |
| | $\text{FAD}_{\text{dec}}\downarrow$ | $\text{KLD}_{\text{dec}}\downarrow$ | $\text{FAD}_{\text{avg}}\downarrow$ | $\text{CC}_\uparrow$ | | | $\text{AUC}_\uparrow$ | | |
| | | | | All | 1fps | 5fps | All | 1fps | 5fps |
|---|---|---|---|---|---|---|---|---|---|
| Directional & Visual | 3.86 | **1.71** | 4.20 | 0.635 | 0.584 | 0.531 | 0.846 | 0.819 | 0.790 |
| Directional only | 5.56 | 2.01 | 4.76 | **0.721** | **0.671** | **0.624** | **0.890** | **0.864** | **0.839** |
| Visual Only | **3.85** | **1.71** | **4.14** | 0.488 | 0.446 | 0.406 | 0.757 | 0.736 | 0.716 |
| Dual 2.5 & 2.5 | 3.93 | 1.72 | 4.22 | 0.549 | 0.505 | 0.459 | 0.795 | 0.772 | 0.748 |
| Dual 7.5 & 2.5 | 4.32 | 1.76 | 4.59 | 0.596 | 0.540 | 0.487 | 0.829 | 0.798 | 0.769 |

## A  IMPLEMENTATION DETAILS OF VISAGE

We utilize a pretrained CLIP model based on ViT-B/32 (Dosovitskiy et al., 2021), where output dimension is 512 and $d_p = 768$. We obtain CLIP embeddings at 4 frames per second (FPS). For the patchwise energy map, the window sizes $N$ and $T$ are both set to 1. For computing the compute energy map $E$ from the scores, a temperature of 0.1 is used for the softmax, and top-$p$ filtering is applied after averaging the probabilities, with a top-$p$ threshold of 0.7. For the DAC, we adopted the 44100 Hz variant, which employs $N = 9$ codebooks per audio channel, each with a size of 1024, producing 86 codes per second of audio.

The transformer architecture have a hidden dimension of $d_t = 1024$ with 16 attention heads. It consistes of 6 layers in the encoder and 12 layers in the decoder. Since the sequence length of visual features is much shorter than that of audio features and adjacent CLIP embeddings are often highly similar, we halve the number of layers in the encoder compared to that of the decoder. Both the energy map $E$ and unit vector $u$ are processed through MLP layers composed of two linear layers with GELU activation in between, using a hidden dimension of 1024. Overall, ViSAGe have 360M trainable parameters.

## B  ABLATION ON CLASSIFIER-FREE GUIDANCE

We conduct an ablation study on different classifier-free guidance schemes. We compare our approach to alternative methods that modify the second term in Eq. 5, including guiding only the visual condition, as in previous video-to-audio generation works (Sheffer & Adi, 2023; Mei et al., 2023), and guiding both conditions separately using dual classifier-free guidance (Lee et al., 2024; Yang et al., 2024), which has been used to improve different aspects of text-to-speech generation. Dual classifier-free guidance assumes that the two input conditions are independent, allowing each condition to be guided separately. However, in our case, visual and directional conditions are closely related, and guiding them separately may not capture their combined effect on spatial audio generation effectively.

For reorganized DAC code matrix $C'_a \in \mathbb{V}^{4N \times (2L_c+1)}$, let $C'_t = (C'_a)_{:,t}$ and $V$ and $D$ respectively denotes the video frames and the camera direction. Each guidance can be formulated as follows. First, for visual only guidance, we sample DAC codes from

$$\log p_\phi(C'_t \mid C'_{1:t-1}, V, D) + \omega(\log p_\phi(C'_t \mid C'_{1:t-1}, V, \varnothing) - \log p_\phi(C'_t \mid C'_{1:t-1}, \varnothing, \varnothing)), \quad (6)$$

where $\omega$ denotes the visual guidance scale and $p_\phi$ denotes the probability parametrized by the ViSAGe model.

Lastly, dual guidance can be formulated as sampling DAC codes from

$$\begin{aligned}
\log p_\phi(C'_t \mid C'_{1:t-1}, V, D) &+ \omega_1(\log p_\phi(C'_t \mid C'_{1:t-1}, \varnothing, D) - \log p_\phi(C'_t \mid C'_{1:t-1}, \varnothing, \varnothing)) \\
&+ \omega_2(\log p_\phi(C'_t \mid C'_{1:t-1}, V, \varnothing) - \log p_\phi(C'_t \mid C'_{1:t-1}, \varnothing, \varnothing)),
\end{aligned} \quad (7)$$

where $\omega_1$ and $\omega_2$ respectively denote the guidance scale for directional guidance and visual guidance.

The results in Table 4 show that omitting directional guidance degrades spatial accuracy, while the absence of visual guidance reduces the semantic quality of the generated audio. Our approach, which

Table 5: Results of the subjective test. "Win" represents the percentage of participants who preferred ViSAGe, "Lose" represents those who preferred the baseline, and "Tie" indicates the percentage of participants with no preference.

| Criteria | Win | Lose | Tie | Win | Lose | Tie |
|---|---|---|---|---|---|---|
| | (a) SpecVQGAN | | | (b) Diff-Foley | | |
| Natural | **43.33** | 25.56 | 31.11 | **44.44** | 14.44 | 41.11 |
| Relevent | **50.00** | 27.78 | 22.22 | **40.00** | 23.33 | 36.67 |
| Spatial | **52.22** | 31.11 | 16.67 | **42.22** | 30.00 | 27.78 |
| Overall | **50.00** | 23.33 | 26.67 | **44.44** | 16.67 | 38.89 |

jointly guides both conditions, outperforms guiding them separately, indicating that the two conditions are closely related. Furthermore, increasing the guidance scale for the directional condition improves the spatial aspect but worsens the semantic quality, and still underperforms compared to guiding both conditions jointly. This provides additional evidence that the two conditions are interdependent and are better guided together rather than guided separately.

## C  TRAINING AND EVALUATION DETAILS

**Training Loss.** For the reorganized DAC code matrix $C'_a \in \mathbb{V}^{4N \times (2L_c+1)}$, given input video frames $V$ and a camera direction $D$, the training loss is formulated as

$$\mathcal{L} = -\frac{1}{(2L_c + 1) \times 4N} \sum_{t=1}^{2L_c+1} \sum_{n=1}^{4N} \log p_\phi \left( (C'_a)_{n,t} \mid (C'_a)_{n,1:t-1}, V, D \right) \quad (8)$$

where $p_\phi$ denotes the probability parametrized by the ViSAGe model.

**Training Hyperparameters.** For pretraining on VGGSound, we use a constant learning rate of 1e-4 with 4000 warmup steps. For finetuning on YT-Ambigen, we apply a constant learning rate of 1e-4 without warmup. When training from scratch on YT-Ambigen, we use a constant learning rate of 2e-4 with 4000 warmup steps. The AdamW optimizer is adopted with a weight decay of 1e-2 and a gradient clipping norm of 1.0. Training is conducted on 2 NVIDIA A6000 or A40 GPUs with a batch size of 64. We also utilize bfloat16 precision and FlashAttention-2 (Dao, 2024) to accelerate the training process.

**Evaluation Details.** For SpecVQGAN, we use a pretrained model based on ResNet-50 (He et al., 2016) features at 21.5 fps, with a total of 310M trainable parameters. SpecVQGAN generates audio at 22050 Hz. For Diff-Foley, we adopt the large variant of the pretrained model, which has 860M trainable parameters and generates audio at 16000 Hz.

For all models, we evaluate the checkpoint with the lowest validation loss. To compute the FAD, we use features from the VGGish network (Hershey et al., 2017) pretrained on AudioSet (Gemmeke et al., 2017) classification. For KLD, we follow previous works (Sheffer & Adi, 2023; Mei et al., 2023) and use the PaSST model (Koutini et al., 2022) pretrained on AudioSet classification, to calculate class distributions. We use the audioldm_eval library (Liu et al., 2023) to compute all metrics. Since these pretrained audio classifiers accept 16000 Hz audio as input, we resampled all generated ambisonics to 16000 Hz before evaluation.

## D  ILLUSTRAION OF CODE GENERATION PATTERNS

The code generation patterns compared in the ablation study in Section 6.2 are illustrated in Figure 4.

## E  SUBJECTIVE TEST RESULTS

We conducted human preference analysis with two-sample hypothesis testing of generated audio with respect to four subjective criteria:

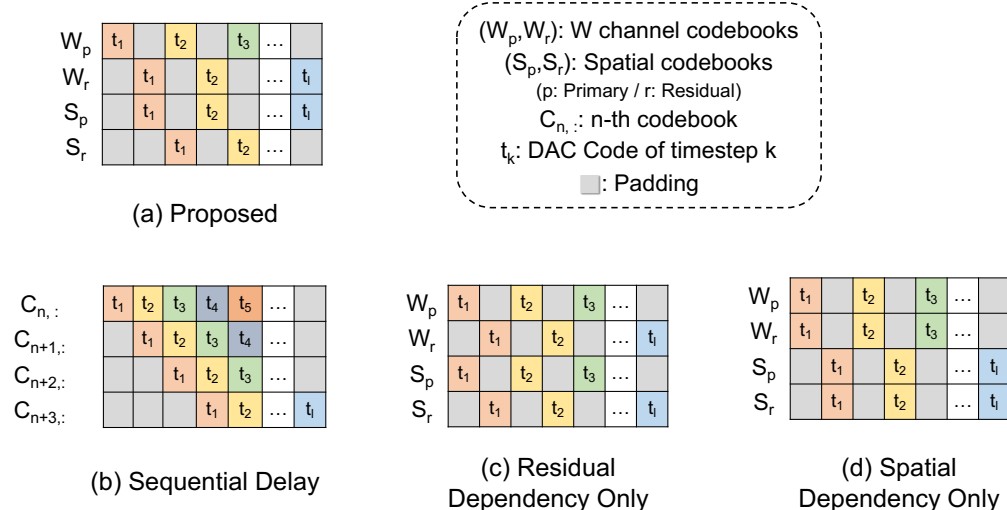

Figure 4: Illustration of different code generation patterns from Section 6.2. For (a), (c), and (d), each block represents all residual codes belonging to the corresponding codebook group. In (b), each block represents a single code from the codebook $C_{n,:}$.

- **Naturalness**: Which audio sounds more natural?
- **Relevance**: Which audio is more closely related to objects and surroundings in the video?
- **Spatiality**: After observing different viewpoints of a 360° video by rotating, which audio better captures the spatial effects perceived in both ears?
- **Overall preference**: Which audio do you prefer overall?

Due to the characteristics of 360° videos and spatial audio, we recruited 12 participants in person instead of crowdsourcing (e.g., MTurk). Each annotator evaluated an average of 15 videos out of 30 randomly selected samples from the test split. The results are summarized in Table 5, showing that our samples are generally preferred over the prior arts across all four criteria. It is worth noting that the gap is particularly large for the spatiality criterion.

## F  ADDITIONAL QUALITATIVE EXAMPLES

**Comparsion to two-stage baselines.** Linear spectrograms generated by ViSAGe and two-stage approaches based on Diff-Foley (Luo et al., 2023) are shown in Figure 5. ViSAGe consistently produces semantically and spatially coherent ambisonics channels. While Diff-Foley generates semantically plausible spectrograms, the spatial channels produced through audio spatialization exhibit limited fidelity and contain several artifacts. We attribute this to two factors: (1) the commonly used mix-and-separate approach based on complex masking tends to introduce artifacts, and (2) the generated audio has limited fidelity compared to ground-truth audio. This worsens the semantic quality of the generated spatial channels by introducing a gap between the training and inference data in spatialization models, highlighting the advantage of our end-to-end approach.

**Ambisonics Generation for Dynamic Visual Scenes.** Figure 6 demonstrates the patchwise energy map of the video frames, along with the audio energy map of the first-order ambisonics generated by ViSAGe. The proposed patchwise energy map effectively highlights dynamic changes in visual scenes. When objects move dynamically within a scene or when specific regions undergo temporal changes, these areas are represented by high energy values due to significant differences with their spatially and temporally neighboring patches.

**Role of Camera Direction.** Figure 7 illustrates the effect of the camera direction parameter on the ambisonics generation. The orientation from which the visual information is captured significantly

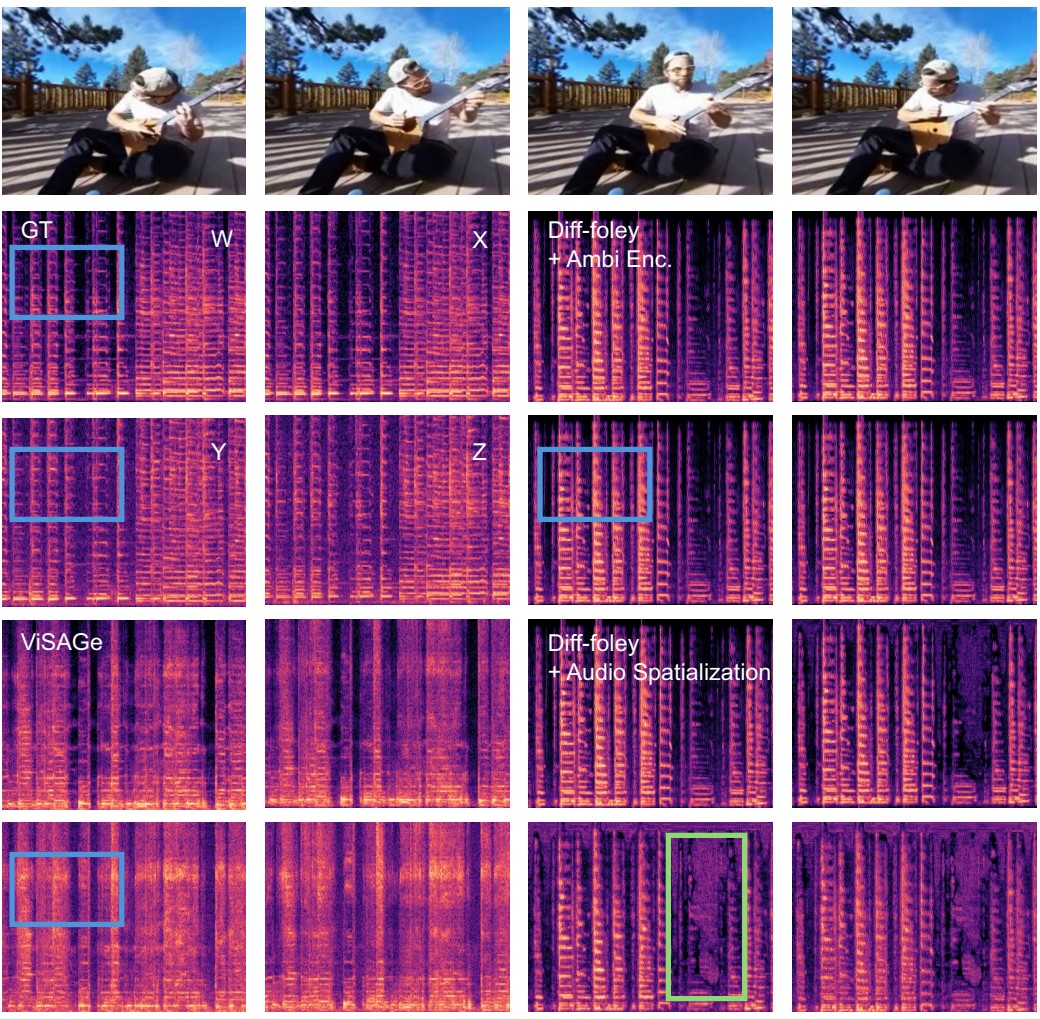

Figure 5: Qualitative examples of generated audios from ViSAGe and two-stage approaches. Blue boxes shows that while ViSAGe captures the acoustic characteristics of surroundings (indicated by less black area in spectrogram) in Y-channel, while Ambi Enc. generates almost identical spectrogram for spatial channels. Green box show that spectrogram of spatial channels generated by Audio Spatialization may introduce artifacts.

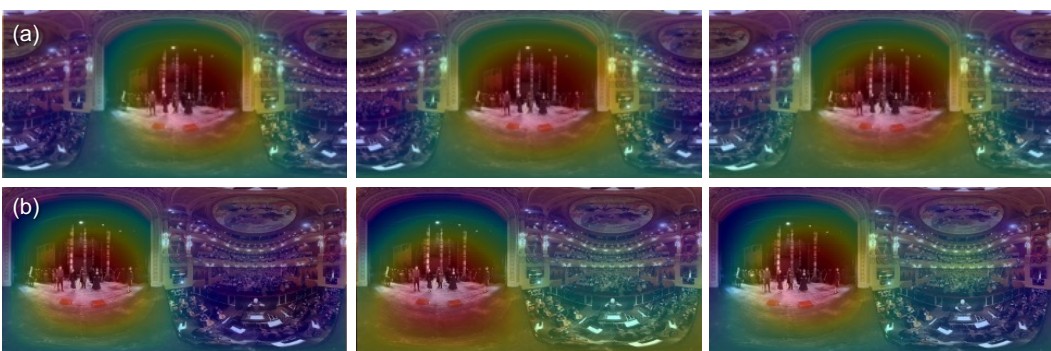

Figure 6: Qualitative example of patchwise energy map and generated audio in rapidly changing scene captured at 4 frames per second.

Figure 7: Qualitative example of the camera direction parameter. (a) Camera direction is set to front: $(0, 0)$. (b) Camera direction is set to front-left: $(\frac{\pi}{3}, 0)$.

impacts the output ambisonics. As described in Sec 3.2, ambisonics capture the full three-dimensional sound field and are commonly used with panoramic videos. However, when paired with a field-of-view (FoV) video, ambiguity arises regarding the visual scene's placement within the three-dimensional space. While treating the FoV scene as a frontal view simplifies processing, it compromises the immersiveness and controllability of ambisonics generation since all sounds appear to originate from directly in front of the listener. To address this, we introduce a camera direction parameter as an additional condition that specifies the visual scene's position within the three-dimensional sound field, enabling proper audio-visual spatial alignment. In practice, the camera direction parameter guides the directivity of spatial audio generation. For instance, in a orchestra recording, if the camera faces front, the audio originates primarily from the front. If the camera turns left, the audio follows, enhancing spatial realism.

More qualitative examples and demo videos are available at https://jaeyeonkim99.github.io/visage

## G   DETAILS OF YT-AMBIGEN

We analyze the content and distribution of the proposed YT-Ambigen dataset. To examine the audio distribution, we classify each audio clip using the PaSST (Koutini et al., 2022) model trained on AudioSet Gemmeke et al. (2017). For video content distribution, we employ an FPN (Lin et al., 2017) to identify the most salient object in each video. Additionally, we utilize CoTracker (Karaev et al., 2025) to capture object motion and trajectories over time.

In Figure 8-(a), our audio distribution is similar to that of AudioSet, where YT-Ambigen covers 314 out of 527 classes in AudioSet, accounting for 97.91% of the entire AudioSet videos. Figure 8-(b-d) summarizes the semantic, spatial, and temporal distributions of the most salient object per video. These objects cover 79 out of 80 classes in COCO and are located in diverse positions within the field of view. These objects would often move around significantly during five-second segments, creating more challenging scenarios for video-to-ambisonics generation.

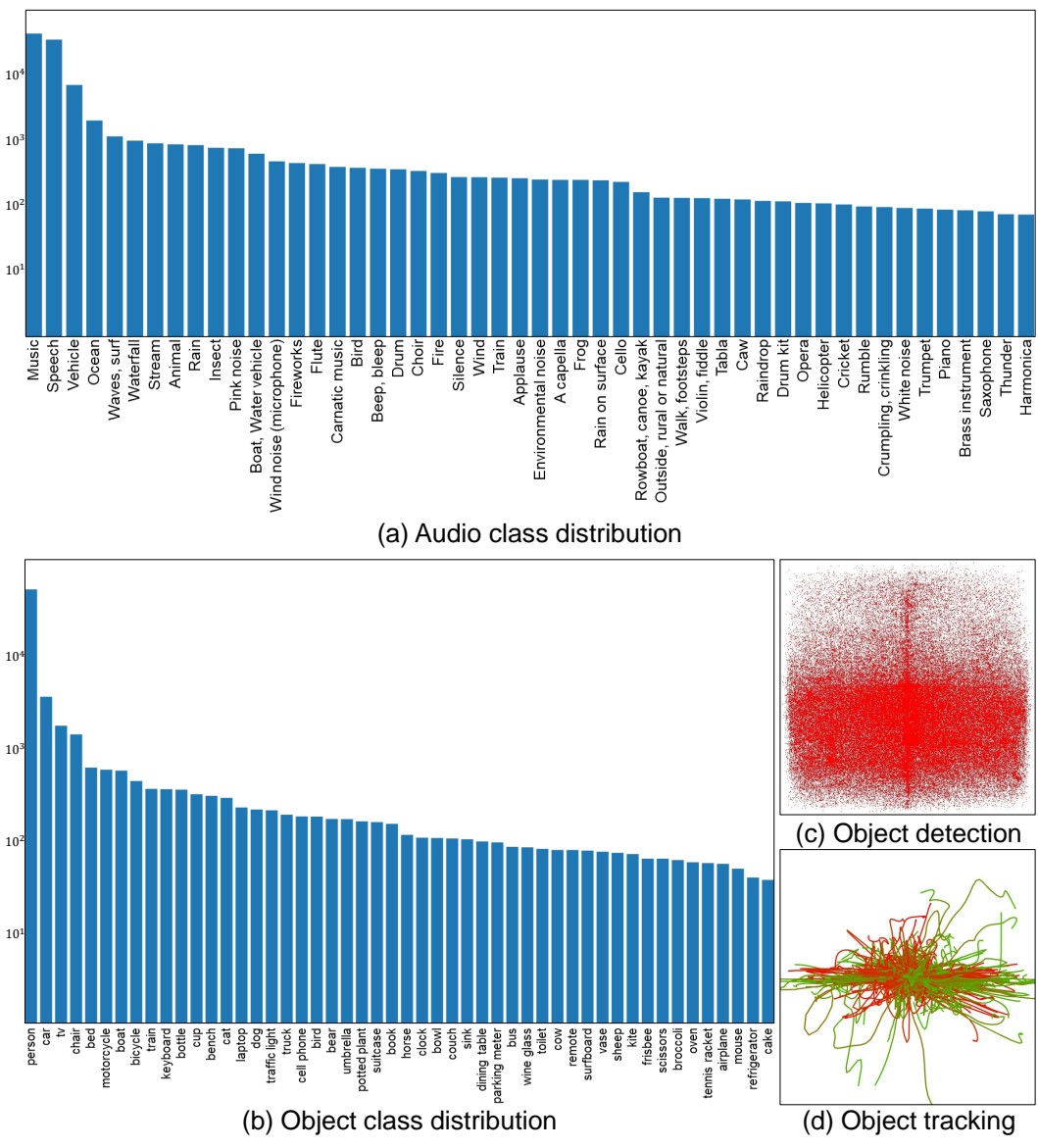

Figure 8: Distribution statistics of YT-Ambigen. (a) The top-50 AudioSet labels distribution predicted with PaSST. (b) The top-50 COCO object class distribution of the most salient object per video with FPN. (c) The center coordinates of each salient object's bounding box. (d) The tracking of center pixels per video predicted with CoTracker (randomly selected 1K samples for visibility).

