# OpenReview forum: "ViSAGe: Video-to-Spatial Audio Generation"
_ICLR.cc/2025/Conference — ICLR 2025 Poster_

### Official Review · Reviewer_vsEj · 2024-11-01

**Soundness:** 3
**Presentation:** 3
**Contribution:** 3
**Rating:** 6
**Confidence:** 4

**Summary:**

The paper introduces ViSAGe, a novel framework for generating first-order ambisonics (a spatial audio format) directly from silent videos. This is significant for enhancing the immersiveness of audio-visual experiences without the need for complex recording systems or specialized expertise.

**Strengths:**

The paper presents a novel approach to generating spatial audio directly from silent videos, aiming to enhance immersion without complex recording setups. A key contribution is the creation of the YT-Ambigen dataset, which includes 102K video clips paired with spatial audio, providing valuable resources for training and evaluation. The authors propose innovative evaluation metrics that utilize audio energy maps and visual saliency to assess the spatial quality of the generated audio, offering a deeper understanding of audio-visual coherence. Their end-to-end framework, ViSAGe, integrates CLIP visual features with neural audio codecs, avoiding issues associated with traditional two-stage approaches. Lastly, this research has significant implications for media production in film, virtual reality, and augmented reality, potentially revolutionizing audio generation for visual content.

**Weaknesses:**

The current approach may not fully capture the dynamic changes in audio that correspond to rapidly changing visual scenes. The generation of spatial audio is based on neural networks, which may not always adhere to the physical principles governing sound propagation and perception.

**Questions:**

Could the authors elaborate on the diversity of the YT-Ambigen dataset in terms of different acoustic environments and video content types? How does this diversity compare to real-world scenarios?

---

> ### Author Response · Authors · 2024-11-23
>
> We sincerely thank Reviewer vsEj for the thoughtful feedback.
>
> ---
>
> ## vsEj-Q1. Diversity of the dataset
>
> Thank you for the suggestion. We included distribution statistics of YT-Ambigen in the Appendix G, which cover:
>
> - (a) The top-50 AudioSet label distribution predicted with PaSST [1]
> - (b) The top-50 COCO object class distribution of the most salient object per video with FPN [2]
> - (c) The center coordinates of each salient object’s bounding box
> - (d) The tracking of center pixels per video predicted with CoTracker [3] (randomly selected 1K samples for visibility).
>
> It is worth noting that the distribution reported in (a) is similar to that of AudioSet, which covers diverse real-world audio events. Among the 527 classes in AudioSet, our YT-Ambigen covers 314 classes, which account for 97.91% of the entire AudioSet videos. Please refer to __T5M6-Q2__ for further explanation.
>
> ---
>
> ## vsEj-Q2. Incorporating dynamic changes in rapidly changing scenes
>
> We have introduced the patchwise energy map to address this challenge, which effectively highlights dynamic changes in visual scenes. When objects move dynamically within a scene or when specific regions undergo temporal changes, these areas are represented by high energy values due to significant differences with their spatially and temporally neighboring patches.
> To provide further clarity, we have added qualitative examples of the patchwise energy map for video frames, alongside audio energy maps for generated audio in the Appendix F.
>
> Additionally, we have expanded our demonstration with more examples featuring dynamically moving scenes on the demo page.
>
> ---
>
> ## vsEj-Q3. Adherence to physical principles
>
> While we recognize the importance of incorporating physical principles, such as room impulse responses (RIRs), to ensure physical plausibility, our task operates in an open-domain setup with diverse and uncontrolled acoustic environments. Explicitly modeling these physical characteristics under such conditions would pose significant challenges.
>
> Instead, we adopted neural networks to implicitly capture acoustic characteristics and approximate physical principles, which are inherently embedded in ambisonics captured in the wild. Despite the lack of explicit enforcement, ViSAGe demonstrated strong adherence to physical principles when compared to baseline methods. For instance, as illustrated in Figure 5, ViSAGe avoided introducing implausible artifacts (as observed in Audio Spatialization) and did not produce nearly identical spectrograms for XYZ channels (as observed in Ambi Enc.).
>
> In future work, we aim to incorporate physical principles more explicitly into our model design and training process. For example, we plan to explore simulated environments such as SoundSpaces [4] to better integrate physical plausibility into spatial audio generation.
>
> ---
>
> [1] Koutini et al. Efficient Training of Audio Transformers with Patchout. In Interspeech 2022.
>
> [2] Lin et al. Feature Pyramid Networks for Object Detection. In CVPR 2017.
>
> [3] Karaev et al. CoTracker: It is Better to Track Together. In ECCV 2024.
>
> [4] Chen et al. SoundSpaces 2.0: A Simulation Platform for Visual-Acoustic Learning. In NeurIPS 2022.

---

> > ### Comment · Reviewer_vsEj · 2024-12-03
> >
> > Thanks for the detailed rebuttal comment. I will maintain my current rating.

---

### Official Review · Reviewer_XV3v · 2024-11-02

**Soundness:** 3
**Presentation:** 3
**Contribution:** 3
**Rating:** 6
**Confidence:** 4

**Summary:**

ViSAGe is a framework designed to generate spatial audio, specifically first-order ambisonics, directly from silent videos, enabling immersive audio experiences without complex equipment. Utilizing the YT-Ambigen dataset of over 102,000 video clips with ambisonic audio, ViSAGe integrates CLIP visual features and a neural audio codec for synchronized, viewpoint-adaptive sound. It outperforms traditional two-stage methods by producing temporally aligned, spatially coherent audio, evaluated through innovative metrics based on audio energy and saliency.

**Strengths:**

1. The paper is well-written
2. The approach of generating spatial audio from FoV holds considerable value for practical applications. Open-sourcing the proposed dataset would provide a valuable resource to the community.
3. The design methodology is reasonable and effective.

**Weaknesses:**

1. There appears to be no explanation of the camera orientation parameter, which leaves me unclear on how it enhances spatial perception.
2. The method assumes that the CLIP visual representation lacks spatial information. Could replacing it with a visual representation that has stronger local perception, such as DINOv2, improve the quality of spatial audio generation?

**Questions:**

See weaknesses

---

> ### Author Response · Authors · 2024-11-23
>
> We deeply appreciate Reviewer XV3v for the helpful suggestion.
>
> ---
>
> ## XV3v-Q1. Role of camera orientation parameter in enhancing spatial perception
>
> The orientation from which the visual information is captured significantly impacts the output ambisonics. As described in Sec 3.2, ambisonics capture the full three-dimensional sound field and are commonly used with panoramic videos. However, when paired with a field-of-view (FoV) video, ambiguity arises regarding the visual scene's placement within the three-dimensional space. While treating the FoV scene as a frontal view simplifies processing, it compromises the immersiveness and controllability of ambisonics generation since all sounds appear to originate from directly in front of the listener. To address this, we introduce a camera orientation parameter as an additional condition that specifies the visual scene's position within the three-dimensional sound field, enabling proper audio-visual spatial alignment.
>
> In practice, the camera orientation parameter guides the directivity of spatial audio generation. For instance, in a orchestra recording, if the camera faces front, the audio originates primarily from the front. If the camera turns left, the audio follows, enhancing spatial realism. To further illustrate this mechanism, we have added qualitative examples of the camera orientation parameter’s effect in Appendix F.
>
> ---
>
> ## XV3v-Q2. Influence of visual representations with stronger local perception (DINOv2) instead of CLIP
>
> The performance of using DINOv2 for visual representations is reported below. We followed the same training procedure explained in Section 5 other than using DINOv2 instead of CLIP. Using DINOv2 is not beneficial to performance for both semantic and spatial metrics. To ensure that the result is not confined to YT-Ambigen, we conducted identical experiments with VGGSound, a well-established benchmark for video-to-audio generation. DINOv2 as visual representations is also detrimental to performance in VGGSound, degrading both FAD (3.62 → 4.33) and KLD (2.23 → 2.25). This performance degradation potentially has to do with differences in dataset scales and pretext tasks during pretraining.
>
> |        | FAD$_\text{dec}$ | KLD$_\text{dec}$ | FAD$_\text{avg}$ | CC$_\text{All}$ | CC$_\text{1fps}$ | CC$_\text{5fps}$ | AUC$_\text{All}$ | AUC$_\text{1fps}$ | AUC$_\text{5fps}$ |
> |--------|:----------------:|:----------------:|:----------------:|-----------------|------------------|------------------|------------------|-------------------|-------------------|
> | CLIP   |       3.74       |       1.77       |       4.04       | 0.524           | 0.482            | 0.439            | 0.778            | 0.757             | 0.734             |
> | DINOv2 |       4.06       |       1.79       |       4.30       | 0.484           | 0.447            | 0.406            | 0.759            | 0.739             | 0.788             |

---

### Official Review · Reviewer_NiUt · 2024-11-03

**Soundness:** 3
**Presentation:** 3
**Contribution:** 3
**Rating:** 6
**Confidence:** 3

**Summary:**

- The paper introduces a new problem of generating spatial audio for silent videos.
- The problem is formalized as:  Given silent video and direction of camera -> generate First Order Ambisonics (FOA)
- Due to lack of suitable datasets, the authors collect YT-AmbiGen from Youtube for this task
- They use discrete audio representations for encoding FOA and autoregressive model for generation
- New baselines and evaluation metrics are proposed to compare and evaluate their approach
- A comprehensive comparison with baselines, along with ablation study, show the effectiveness of their proposed approach

**Strengths:**

- The problem is new, interesting, and important for the research community
- The authors introduce a new dataset for this new task, setting a benchmark for future spatial audio generation research.
- Suitable baselines and metrics are proposed to compare on this new task
- Carefully designed components are incorporated, for eg. FOA encoding, sequence of its generation, rotation augmentation, patchwise energy maps

**Weaknesses:**

Major:

Missing Subjective tests:
- The paper lacks subjective evaluations; studies assessing quality, and directionality ( or localization accuracy) should be included. Authors should compare their approach with baselines on metrics like mean opinion score (or other subjective metrics).

Demo examples
- While the demo examples appear semantically good, the sounds are often too diffuse, making it challenging to precisely localize the direction of the audio.
- Including some static sources with smaller, more focused sound-generation areas can help in better experiencing and analyzing the sound source direction (subjectively).

Minor:
- Missing relevant references:

Some recent work on spatial audio generation should be referenced. These methods also generate spatial audio (FOA) given some conditions (eg. direction of arrival, and sound source category).

[1] Heydari et. al, "Immersediffusion: A generative spatial audio latent diffusion model"

[2] Kushwaha et. al, "Diff-SAGe: End-to-End Spatial Audio Generation Using Diffusion Models"

**Questions:**

- Could the authors include subjective test results evaluating metrics such as audio quality and relevance to the specified direction?

- Would it be possible to create a subset of clean, single-source static sounds as a benchmark and demo set? This would enable evaluation using metrics like Direction of Arrival (DoA) and provide clearer, more focused demo examples.

---

> ### Author Response · Authors · 2024-11-23
>
> We greatly appreciate Reviewer NiUt for the constructive feedback.
>
> ---
>
> ## NiUt-Q1. Subjective test results
>
> Thank you for the feedback. We conducted human preference analysis with two-sample hypothesis testing of generated audio with respect to four subjective criteria:
>
> - __Naturalness__: Which audio sounds more natural?
> - __Relevance__: Which audio is more closely related to objects and surroundings in the video?
> - __Spatiality__: After observing different viewpoints of a 360° video by rotating, which audio better captures the spatial effects perceived in both ears?
> - __Overall preference__: Which audio do you prefer overall?
>
> Due to the characteristics of 360° videos and spatial audio, we recruited 12 participants in person instead of crowdsourcing (e.g., MTurk). Each annotator evaluated an average of 15 videos out of 30 randomly selected samples from the test split. The results are summarized below, showing that our samples are generally preferred over the prior arts across all four criteria. It is worth noting that the gap is particularly large for the spatiality criterion.
>
> |(a)SpecVQGAN|Win|Tie|Lose|
> |---|:-:|:-:|:-:|
> |Natural |43.33|25.56|31.11|
> |Relevant|50.00|27.78|22.22|
> |Spatial |52.22|31.11|16.67|
> |Overall |50.00|23.33|26.67|
>
> |(b)Diff-Foley|Win|Tie|Lose|
> |---|:-:|:-:|:-:|
> |Natural |44.44|14.44|41.11|
> |Relevant|40.00|23.33|36.67|
> |Spatial |42.22|30.00|27.78|
> |Overall |44.44|16.67|38.89|
>
> ---
>
> ## NiUt-Q2. Curating a clean subset of test split
>
> Thank you for the suggestion. We have applied a tighter set of conditions on the test split of YT-Ambigen to select 1.5K samples (i.e., about 15% of the test split) with improved cleanliness as _mini-test_:
>
> - Higher CAVP and PaSST filtering thresholds for improved audio-visual correspondence and audio event likelihood, respectively
> - Sampling ambisonics with clearer directivity by measuring inter-channel correlation.
>
>
> Using the mini-test set improves KLD (1.71 → 1.48), CC (0.635 → 0.718 ), and AUC (0.846 → 0.897) at the cost of FAD (3.86 → 6.10). This implies that a distribution shift may occur during the selection of the semantically localized subset, as the sound becomes less diffused, thereby enhancing the directivity-related metrics. We believe this mini-test split will also be useful for analyzing other aspects of the benchmark. As such, we will publicly release the mini-test split.
>
> ---
>
> ## NiUt-Q3. More intuitive qualitative examples
>
> We included qualitative examples from the clean subset explained above in our demo page.
> Please refer to __T5M6-Q3__ for additional qualitative examples.
>
> ---
>
> ## NiUt-Q4. Missing relevant references
>
> Both arXiv papers were released in late October, which is after the ICLR submission deadline. A key difference from our approach is that they use synthesized spatial audio for conditional audio generation, while we leverage real ambisonics captured in the wild. We will add these references in our final draft.

---

> > ### Comment · Reviewer_NiUt · 2024-11-24
> >
> > I would like to thank the authors for diligently answering my questions. I have raised my score.
> >
> > I have an additional question, did authors consider diffusion approaches for this task? If yes, what benefit does autoregressive approach (in spatial audio generation) provide over diffusion approaches?

---

> > > ### Author Response · Authors · 2024-11-26
> > >
> > > We sincerely appreciate the reviewer’s supportive feedback.
> > >
> > > We initially considered both approaches for this task. Specifically, in the context of the widely used latent diffusion method, two primary approaches appear viable for audio generation: (i) generating the latents for the four channels independently, or (ii) concatenating the latents of all four channels and generating them together. However,  generating channels separately or using a concatenated representation could require approximately four times more computation than the mono channel generation and may struggle to explicitly capture dependencies between channels. In contrast, we believe that designing a well-structured code generation pattern for an autoregressive approach, as demonstrated in our extensive experiments and Xpfx-Q1, is both computationally more efficient and better suited for modeling inter-channel dependencies.

---

> ### Comment · Reviewer_NiUt · 2024-11-27
> **Regarding demo videos coming from train split**
>
> I agree with the public comment (by Yang Liu) regarding the demo videos coming from the training set (I also checked the split), which may indicate overfitting of the model and a lack of generalizability to the test set.
> Demos are important for generative tasks, and given this concern, I would lower my score but still remain positive due to the promising direction of the work and the presence of a few good demo examples from the test split.

---

### Official Review · Reviewer_T5M6 · 2024-11-03

**Soundness:** 3
**Presentation:** 3
**Contribution:** 3
**Rating:** 6
**Confidence:** 3

**Summary:**

This paper presents a novel task: generating spatial audio from silent video. Specifically, given a silent video and the camera direction, the proposed model, ViSAGe, leverages CLIP features, patchwise energy maps, and neural audio codes, incorporating a code generation scheme to simultaneously produce multiple spatial channels of audio in the first-order ambisonic format. For this task, the authors introduce a new dataset, YT-Ambigen, which consists of YouTube videos paired with first-order ambisonics. Compared to two-stage models, which first generate mono-channel audio and then perform audio spatialization, the proposed ViSAGe outperforms in overall quantitative metrics.

**Strengths:**

- The paper is clearly written and well-presented.
- The proposed task of generating spatial audio from silent video is interesting and takes one step further of existing works that tackle this task separately.
- The dataset curation pipeline effectively addresses the limitations of existing datasets, and the dataset would make contribution to the community.

**Weaknesses:**

- Lack of moving object samples or objects that are not centered.
    - When listening to the synthesized audio while watching the video, most content appears centered, which makes the synthesis task relatively simple and appears to be similar to the mono audio generation task.
    - To fully demonstrate the effectiveness of the proposed method and task, more qualitative examples are needed that show results in challenging scenarios (e.g., objects moving from left to right, objects not centered, or visual events requiring time-synced audio synthesis).
- This lack raises questions about dataset quality.
    - Since the dataset seems to be collected automatically, is the test set clean or challenging enough to serve as a reliable benchmark? Wouldn’t human annotation or verification be necessary to validate its quality?
    - Furthermore, what distribution does this dataset cover? What types of events or objects appear in it? If the task involves not only audio spatialization but also requires semantic information, then details on the dataset (e.g., statistics, categories) should be provided.
- Clearer visualization and explanation in qualitative results would enhance the work.
    - The generated spectrograms in Fig. 3(a) appear different from the ground truth. If the authors do not highlight the specific areas readers should focus on, important details may be overlooked.
    - Similarly, in Fig. 3(b), overlapping the heatmap with the original video would improve clarity. While the predicted heatmap seems better than the baseline model, it still does not fully align with the real visual events.

**Questions:**

- Two numbers are highlighted in bold in Table 3: Ablation on Model Components.
- Please check that all figure and table references are correctly cited, e.g., L476 references Figure D.
- Could the authors clearly describe how the 9 codes per timestep in each audio channel are generated? The figures seem to illustrate the process for generating a single code, while the neural codec contains 9 RVQ codes per timestep.

---

> ### Author Response · Authors · 2024-11-23
>
> We sincerely thank Reviewer T5M6 for the thoughtful feedback and valuable suggestions.
>
> ---
>
> ## T5M6-Q1. Quality control of the dataset
>
> We established the reliability of our dataset's quality in two aspects. First, the quality of audio generated using YT-Ambigen is in line with well-established large-scale benchmarks, e.g., VGGSound. As reported in Table 2, the FAD (3.95 vs. 3.62) and KLD (1.77 vs. 2.23) scores of our model trained with YT-Ambigen's mono channel are similar to those of VGGSound. These metrics from our VGGSound-trained model are also comparable to the prior arts like SpecVQGAN and Diff-Foley, suggesting that YT-Ambigen offers a reliable data source for benchmarking video-to-audio generation.
>
> Second, using a combination of performant off-the-shelf multimodal discriminators for filtering is a well-established practice for constructing large-scale datasets with quality control [1, 2, 3], and could be more effective than manual filtering in some cases. For instance, in Table 2, our predecessor in discriminative audio-visual reasoning (YT360) reports an FAD metric of 15.91 for video-to-audio generation, despite being collected through manual filtering.
>
> ---
>
> ## T5M6-Q2. Distribution statistics of the dataset
>
> Thank you for your suggestion. We included the distribution statistics of YT-Ambigen in the Appendix G, which cover:
>
> - (a) The top-50 AudioSet label distribution predicted with PaSST [1]
> - (b) The top-50 COCO object class distribution of the most salient object per video with FPN [2]
> - (c) The center coordinates of each salient object’s bounding box
> - (d) The tracking of center pixels per video predicted with CoTracker [3] (randomly selected 1K samples for visibility).
>
> Our audio distribution is similar to that of AudioSet, where YT-Ambigen covers 314 out of 527 classes in AudioSet, accounting for 97.91% of the entire AudioSet videos. Moreover, the semantic, spatial, and temporal distributions of the most salient object per video are summarized in (b-d). These objects cover 79 out of 80 classes in COCO. Moreover, they are located in diverse positions within the field of view and often move around during five-second segments, creating more challenging scenarios for video-to-ambisonics generation.
>
> ---
>
> ## T5M6-Q3. More examples covering challenging scenarios
>
> We included more qualitative examples with non-centered or moving objects in our demo page and Appendix F.
>
> ---
>
> ## T5M6-Q4. Codebook generation with nine residual codebooks
>
> We apologize for the confusion. As shown in the legend of Figure 2, each block represents a codebook group rather than an individual code. All residual codes from the selected groups are generated at each sequence step. For example, with 9 RVQ codes per channel:
>
> - Step 1: 1 code from $W_p$
> - Step 2: 8 codes from $W_r$ and 3x1 from $S_p$ (total: 11)
> - Step 3: 1 code from $W_p$ and 3x8 from $S_r$ (total: 25)
> - and so on.
>
> The figure caption has been updated to clarify this process.
>
> ---
>
> ## T5M6-Q5. Typos and visualization suggestions
>
> Thank you for your suggestions. We fixed the typos and updated visualizations in Figure 3 and 5.
>
> ---
>
> [1] Lee et al. Automatic Curation of Large-Scale Datasets for Audio-Visual Video Representation Learning. In ICCV 2021.
>
> [2] Nagrani et al. Learning Audio-Video Modalities from Image Captions. In ECCV 2022.
>
> [3] Wang et al. A Large-scale Video-Text Dataset for Multimodal Understanding and Generation. In ICLR 2024.

---

> > ### Comment · Reviewer_T5M6 · 2024-11-25
> >
> > Dear authors,
> >
> > I greatly appreciate your responses and hard work on paper revision.
> > Most of my concerns have been resolved. Thank you so much.
> >
> > Best, Reviewer T5M6

---

> ### Author Response · Authors · 2024-11-26
>
> Dear Reviewer T5M6,
>
> We deeply appreciate your supportive feedback.
>
> Best regards,
>
> Authors.

---

### Official Review · Reviewer_Xpfx · 2024-11-03

**Soundness:** 3
**Presentation:** 3
**Contribution:** 3
**Rating:** 8
**Confidence:** 3

**Summary:**

The paper titled "ViSAGe: Video-to-Spatial Audio Generation" introduces a novel framework for generating first-order ambisonics, a spatial audio format, directly from silent video clips. The authors address the challenge of enhancing the immersiveness of audio-visual experiences without the need for complex recording systems or specialized expertise. They present YT-Ambigen, a dataset of 102K YouTube video clips paired with first-order ambisonics, and propose new evaluation metrics based on audio energy maps and saliency metrics. The core of their work is the Video-to-Spatial Audio Generation (ViSAGe) framework, which leverages CLIP visual features and autoregressive neural audio codec modeling to generate spatial audio that is both semantically rich and spatially coherent. The framework outperforms two-stage approaches and demonstrates the ability to adapt to dynamic visual contexts, showing potential for applications in immersive media production.

**Strengths:**

1. The paper introduces a groundbreaking approach to directly generate spatial audio from video, addressing a previously unsolved problem and offering a significant advancement in the field of immersive media.
2. The ViSAGe framework is an end-to-end solution that integrates neural audio codecs with visual features, which is a novel combination in the context of audio generation from video.
3. The paper is well-structured, with a clear problem statement, including the introduction of a new dataset and evaluation metrics, which are crucial for the field.
4. The creation of YT-Ambigen dataset and new evaluation metrics shows a comprehensive approach to both generating and validating the spatial audio.

**Weaknesses:**

As shown in Figure 2, the framework designed in this paper uses many different modules. Therefore, the computational complexity (model parameters) and running time (inference time) of the overall framework need to be discussed.

**Questions:**

The audio energy map shown in Figure 3 seems to be highly correlated with the location of the sound source, which can be related to the sound source localization task. In other words, can the relevant design ideas under the sound source localization task provide some guidance here? More specifically, can the cross modal contrastive learning strategy [1] commonly used in sound source localization tasks be applied here to impose some additional constraints?

[1] Chen H, Xie W, Afouras T, et al. Localizing visual sounds the hard way[C]//Proceedings of the IEEE/CVF conference on computer vision and pattern recognition. 2021: 16867-16876.

---

> ### Author Response · Authors · 2024-11-23
>
> We deeply appreciate Review Xpfx’s constructive feedback.
>
> ---
>
> ## Xpfx-Q1. Space-time complexity analysis
>
> The number of parameters and the inference time for each model are summarized below. Inference time is computed end-to-end for 320 samples using a batch size of 32, including all auxiliary computations to generate outputs like the Griffin-Lim Algorithm in Diff-Foley [1]. Our framework is up to 1.7x faster than prior arts while using similar or fewer parameters, presumably due to fewer decoder layers and the vocoder-less design.
>
> |Model|Trainable Parameters(M)|Overall Parameters(M)|Inference Time(s/it)|
> |--------------|--------------------------|-------------------------------------------------------------|-----------------------|
> |SpecVQGAN|307.0|ResNet50 (23.5) + Transformer (307.0) + VQGAN Decoder (42.8) + MelGAN (4.3)|3.830|
> |Diff-Foley|859.5|CAVP (32.7) + LDM (859.5) + Latent Decoder (49.5) + Guidance Classifier (11.7)|3.996|
> |Spatializer|80.5|CLIP (87.9) + U-Net (80.5)|0.043|
> |ViSAGe|358.6|CLIP (87.9) + Transformer (358.6) + DAC Decoder (56.5) |2.289|
>
> ---
>
> ## Xpfx-Q2. Adapting cross-modal contrastive learning as in VGG-SS
>
> To perform cross-modal contrastive learning in source localization tasks as in VGG-SS, the model requires ground-truth audio inputs, which are not available in video-to-audio generation tasks. One conceivable strategy to utilize knowledge from cross-modal contrastive learning (e.g., [1] or [2]) without ground-truth audio is to apply reranking among generated audio candidates, i.e., selecting the candidate with highest score as a positive. Model performance under diverse backbones are reported below. Although it takes several times longer to generate outputs, the quantitative metrics remain virtually unchanged with reranking among 10 candidates.
>
> |               | FAD$_\text{dec}$ | KLD$_\text{dec}$ | FAD$_\text{avg}$ | CC$_\text{All}$ | CC$_\text{1fps}$ | CC$_\text{5fps}$ | AUC$_\text{All}$ | AUC$_\text{1fps}$ | AUC$_\text{5fps}$ |
> |---------------|:----------------:|:----------------:|:----------------:|-----------------|------------------|------------------|------------------|-------------------|-------------------|
> | w/o Rerank    |       3.86       |       1.71       |       4.20       | 0.635           | 0.584            | 0.531            | 0.846            | 0.819             | 0.790             |
> | Rerank (CAVP [1]) |       3.85       |       1.71       |       4.25       | 0.635           | 0.581            | 0.526            | 0.846            | 0.818             | 0.788             |
> | Rerank (FNAC [2]) |       3.85       |       1.70       |       4.26       | 0.632           | 0.580            | 0.526            | 0.845            | 0.817             | 0.788            |
>
> ---
>
> [1] Luo et al., Diff-Foley: Synchronized Video-to-Audio Synthesis with Latent Diffusion Models. In NeurIPS 2023.
>
> [2] Sun et al., Learning Audio-Visual Source Localization via False Negative Aware Contrastive Learning. In CVPR 2023.

---

### Author Response · Authors · 2024-11-23
**Response to all reviewers**

We thank the reviewers for their helpful feedback. We appreciate that they acknowledge our effective methodology for ambisonics generation (Xpfx, NiUt, XV3v, vsEj) and our clear writing (Xpfx, T5M6, XV3v). Most importantly, they find our novel dataset and framework provide a meaningful contribution to the research community and relevant applications (Xpfx, T5M6, NiUt, XV3v, vsEj).

We have uploaded the revised paper, with all modified components highlighted in magenta. Additionally, we have addressed the questions raised by each reviewer in our responses. Please note that our dataset, YT-Ambigen, is made available through the link provided in our paper. We will incorporate the feedback and make the official release of the dataset publicly accessible.

---

### Public Comment · ~Yang_Liu131 · 2024-11-27
**Concern Regarding the Use of Training Data in Demo Page of Submission**

Dear Area Chairs, Reviewers, Authors, and Community Members,

I am an active participant in the audio community, and during my reimplementation of this paper, I have encountered an issue that requires your attention. Upon reviewing the demo page associated with this submission, I noticed that a significant portion of the examples used for demonstration purposes appear to be directly sourced from the provided training dataset ('train.csv').

Specifically, my observations include:
1. **Demo Videos Section**: All instances, including BigJ9LPLEcg_464, 5Rj8tpOTonQ_49, aNMEY_wK_O4_157, and 22kR2g5KWYA_40, seem derived from the training data.
2. **Comparison with Baselines Section**: The example "22kR2g5KWYA_40" is also present in the training dataset.

Utilizing training data in demonstrations can potentially lead to misleading conclusions about the model's performance, as it may not accurately reflect the model's generalization capabilities on new, unseen data, thus risking overfitting.

I respectfully urge the committee to examine this issue closely to ensure fair and accurate reporting of the results.

---

> ### Comment · Area_Chair_eco1 · 2024-11-27
>
> Dear Authors,
>
> Could you please address the following concern: Is it true that a significant portion of the examples used for demonstration purposes were directly sourced from the training dataset? If examples from the training set were used in the demo, it could raise concerns about the reliability of the results and the model's true capabilities.
>
>
> Best,
>
> Area Chair

---

> ### Author Response · Authors · 2024-11-27
>
> Dear Yang Liu,
>
> Thank you for your interest and reporting this issue! Except for the four mistakenly included samples, we want to reassure that all our experimental analysis and conclusions in our paper are indeed fair and accurate, which are manifested through multiple demo examples that were unseen during training and through extensive experiments conducted under rigorous conditions. After thorough inspection of all qualitative samples, we confirmed that 4 out of 16 samples on the demo page were from train split and replaced these with additional examples from the validation or test splits.
>
> Best Regards,
> Authors

---

> > ### Public Comment · ~Yang_Liu131 · 2024-11-28
> > **Clarification and Further Inquiry on Sample Origin**
> >
> > Dear Authors,
> >
> > Thank you for your prompt response regarding the issue I raised. Upon further examination, I found that a total of five samples, rather than four as you mentioned, were from the training set. Additionally, I noticed that all the Demo Videos presented in the first chapter are indeed samples from the training dataset. It seems unusual for the model to generate examples that are already present in the training set.
> >
> > Moreover, it's perplexing that this issue was not identified during the post-processing phase, such as when combining audio and video tracks, or even during the upload to YouTube. This raises further questions for me as to why this was overlooked.
> >
> > I appreciate your efforts in addressing these concerns and look forward to any additional clarifications you can provide.

---

> > > ### Author Response · Authors · 2024-11-28
> > >
> > > Dear Yang Liu,
> > >
> > > Thank you for your follow-up and for further examining the issue. First, we would like to clarify that the four samples from the training set—BigJ9LPLEcg_464, 5Rj8tpOTonQ_49, aNMEY_wK_O4_157, and two identical samples of 22kR2g5KWYA_40—correspond to the five samples you mentioned. Additionally, the sample c4iZQAsp088_103, which is part of the test split, was included in the first chapter (Demo Videos) in the initial version.
> > >
> > > To be transparent, we acknowledge that a mistake was made when selecting the demo examples. While conducting experiments with various setups to improve the dataset, we generated numerous samples and kept representative ones together for internal analysis. Unfortunately, when selecting demo videos, we overlooked the official splits and included training samples for the demo videos. We ensure that this mistake in demo selection did not impact the experimental validation at all.
> > >
> > >
> > > We sincerely apologize for this oversight and understand the concerns it raises. To address this, we have excluded training samples and uploaded new demo samples from the validation and test sets, which we hope will resolve your concerns.
> > >
> > > Best Regards, Authors

---

### Meta-Review · Area_Chair_eco1 · 2024-12-21

**Metareview:**

This paper addresses the challenging problem of visually guided spatial audio generation. It introduces ViSAGe, an end-to-end framework that generates first-order ambisonics (FOA) audio directly from silent videos. To support this task, the authors created YT-Ambigen, a large dataset of 5-second YouTube clips paired with FOA recordings, and proposed novel evaluation metrics based on energy maps and saliency cues to measure spatial alignment.

All five reviewers provided positive ratings, with one rating it 8 and four rating it 6 (above threshold). They recognized the novel approach of generating FOA from videos, the thorough experiments, and the new dataset. The concerns in public comments about using training samples in demos were addressed by removing those examples. The authors also clarified that this did not affect the reported results in the main paper. Given the problem's novelty, the dataset, and the results demonstrating improvements over two-stage baselines, I recommend accepting this paper. The authors should revise the paper by incorporating their clarifications and experimental results.

**Additional Comments On Reviewer Discussion:**

The interactive review process, containing both the reviewers' insightful comments and the authors' responsive actions, substantially enhanced the clarity of the paper, the rigor of the experimental results, and the presentation of the dataset. The most significant concern, regarding the use of training data in the demo samples, was addressed with transparency and did not impact the paper's core findings.  Additional experiments, detailed clarifications, and further dataset information can well validate the paper's contributions.  The authors were advised to integrate all clarifications and new results into the final version of the manuscript.

---

### Decision · Program_Chairs · 2025-01-22

Accept (Poster)